# Detecting layer height of smoke aerosols over vegetated land and water surfaces via oxygen absorption bands: Hourly results from the EPIC/DSCOVR in deep space

Xiaoguang Xu[1,2], Jun Wang[1], Yi Wang[1], Jing Zeng[1], Omar Torres[3], Jeffrey S. Reid[4], Steven D. Miller[5], J. Vanderlei Martins[2], Lorraine A. Remer[2]

[1]Department of Chemical and Biochemical Engineering, Center for Global and Regional Environmental Research, and Informatics Initiative, The University of Iowa, Iowa City, Iowa, 52241, USA
[2]Joint Center for Earth Systems Technology and Department of Physics, University of Maryland Baltimore County, Baltimore, Maryland, 21250, USA
[3]Atmospheric Chemistry and Dynamics Laboratory, NASA Goddard Space Flight Center, Greenbelt, Maryland, 20770, USA
[4]Marine Meteorology Division, Naval Research Laboratory, Monterey, California, 93943, USA
[5]Cooperative Institute for Research in the Atmosphere, Colorado State University, Fort Collins, Colorado, 80523, USA

*Correspondence to*: Jun Wang (jun-wang-1@uiowa.edu); Xiaoguang Xu (Xiaoguang-xu@uiowa.edu)

**Abstract.** We present an algorithm for retrieving aerosol layer height (ALH) and aerosol optical depth (AOD) for smoke over vegetated land and water surfaces from measurements of the Earth Polychromatic Imaging Camera (EPIC) onboard the Deep Space Climate Observatory (DSCOVR). The algorithm uses Earth-reflected radiances in six EPIC bands in the visible and near-infrared and incorporates flexible spectral fitting that accounts for specifics of land and water surface reflectivity. The fitting procedure first determines AOD using EPIC atmospheric window bands (443 nm, 551 nm, 680nm, and 780 nm), then uses oxygen ($O_2$) A and B bands (688 nm and 764 nm) to derive ALH, which represents an optical centroid altitude. ALH retrieval over vegetated surface primarily takes advantage of measurements in the $O_2$ B band. We applied the algorithm to EPIC observations of several biomass burning events over the United States and Canada in August 2017. We found that the algorithm can be used to obtain AOD and ALH multiple times daily over water and vegetated land surface. Validation is performed against aerosol extinction profiles detected by the Cloud-Aerosol Lidar with Orthogonal Polarization (CALIOP) and against AOD observed at nine Aerosol Robotic Network (AERONET) sites, showing, on average, an error of 0.58 km and a bias of -0.13 km in retrieved ALH and an error of 0.05 and a bias of 0.03 in retrieved AOD. Additionally, we show that the aerosol height information retrieved by the present algorithm can potentially benefit the retrieval of aerosol properties from EPIC's ultraviolet (UV) bands.

## 1 Introduction

Aerosol vertical distribution is an important but poorly constrained variable that strongly influences how aerosol particles affect earth's energy budget. In particular, absorption of solar radiation by smoke and dust particles can result in diabatic heating, alter atmospheric stability, and affect cloud formation and life cycle. These effects depend critically on the altitude of aerosol layers (Babu et al., 2011; Ge et al., 2014; Koch and Del Genio, 2010; Satheesh et al., 2008; Wendisch et al., 2008). Consequently, aerosol profile factors into the magnitude and even the sign of aerosol direct and indirect effects. An accurate representation of aerosol altitude is thus essential for model prediction of weather and climate (Choi and Chung, 2014; Samset et al., 2013). The thermal signature of dust, in particular, can likewise influence the Earth's longwave budget, and through the interference of retrievals of water vapor and temperature, influence measurement of atmospheric state

(Maddy et al., 2012). Additionally, the knowledge of ALH is essential for retrieving aerosol absorption properties in the ultraviolet (UV) channels (Torres et al., 1998), aerosol microphysical properties from multi-angular photopolarimetric measurements (Chowdhary et al., 2005; Waquet et al., 2009), and for atmospheric correction for ocean color remote sensing (Duforêt et al., 2007).

Despite the importance of aerosol vertical distribution, the simulation of aerosol layer height (ALH) in current climate models is subject to large inter-model variation and uncertainty (Kipling et al., 2016; Koffi et al., 2012). Although the assimilation of space-borne Lidar (e.g., CALIOP, or the Cloud-Aerosol Lidar with Orthogonal Polarization) observations can improve the prediction of vertical allocation (Zhang et al., 2011), a significant challenge remains due to the sparsity of lidar observations over space and time. Therefore, frequent satellite observations of global aerosol vertical distribution based on

more abundant observation sources are critically needed.

One such source, the recently-launched Deep Space Climate Observatory (DSCOVR) mission, has introduced an unprecedented opportunity to acquire ALH information multiple times daily. Launched on 11 February 2015, the DSCOVR spacecraft flies a Lissajous orbit around the Sun-Earth $L_1$ Lagrangian point about 1.5 million km from Earth in the direction of the Sun. While its primary payload is the Plasma-Magnetometer that measures solar wind, DSCOVR also carries the Earth

Polychromatic Imaging Camera (EPIC) to image the sunlit disk of Earth every $60 - 100$ minutes from its stable $L_1$ vantage point. EPIC captures its imagery at a 2048-by-2018-pixel resolution, resolving about 8-km pixel resolution at the disk's center. EPIC was equipped with a double wheel filter to sequentially measure Earth-reflected solar radiance in ten narrow bands spanning from the UV to the near-infrared (NIR) spectrum (Figure 1a), with central wavelengths at 318 nm, 325 nm, 340 nm, 388 nm, 443 nm, 551 nm, 680 nm, 688 nm, 764 nm, and 780 nm (Marshak et al., 2018).

As shown in Figures 1b-c, two of EPIC's bands are located within the oxygen ($O_2$) "A" and "B" bands (764 nm and 688 nm), each associated with a reference continuum band at 780 nm and 680 nm, respectively. These four bands, offering spectral contrasts between absorption bands and continuum bands (known as the Differential Optical Absorption Spectroscopic, or DOAS, ratios), were originally designed for determining cloud height (Yang et al., 2013). Recently, Xu et al. (2017) presented an algorithm to simultaneously retrieve aerosol optical depth (AOD) and ALH using the EPIC

measurements via these four bands and, for the first time, demonstrated EPIC's promising application for determining dust plume height over ocean surfaces during the daytime hours.

The present study, building upon the development of Xu et al. (2017) for determining dust ALH over ocean, extends the algorithm to retrieve ALH from EPIC measurements over land surfaces as well. The new developments in this study includes implementing smoke aerosol optical properties, land surface characterization, and more robust strategies for the

procedures of pixel selection and spectral fitting. The augmentation of the Xu et al. (2017) algorithm takes an important additional step towards our goal of providing more frequent global ALH and AOD information for multi-species global aerosol.

The paper is outlined as follows. Section 2 briefly reviews the physical principle for remote sensing of ALH in the $O_2$ absorption bands and explains the challenges for retrieving ALH over land surfaces, limiting our algorithm development to water and vegetated land surface. Section 3 describes the key elements, procedures, and assumptions of the updated ALH retrieval algorithm. ALH retrievals smoke events over Canada and the United States in August 2017 are demonstrated in

Section 4. Section 5 evaluates the retrieved smoke ALH and AOD from EPIC against aerosol extinction profiles measured by CALIOP and AOD measured at AERONET sites. Section 6 summarizes our findings and discusses potential broader applications of this new ALH information.

## 2 Remote sensing principle and challenges

The ALH retrieval from EPIC takes advantage of the absorption features of molecular $O_2$ in the A band at 759 – 771 nm and the B band at 686 – 695 nm. The electronic transitions from the ground state to two excited states of an $O_2$ molecule, coupled with vibrational-rotational transitions, produce $O_2$ absorption lines in the A and B bands with distinct spectral variability (Figures 1b-c). As a result, the large variability of atmospheric opacity within the $O_2$ A and B bands leads to a wide range of penetration depths of solar radiation (Ding et al., 2016). The spectroscopic characteristics in the $O_2$ bands are related to how

scattered light from aerosol particles interacts with $O_2$ absorption at different altitudes. Furthermore, since $O_2$ molecules are well mixed in the atmospheric column, the altitude-dependence of $O_2$ absorption can provide information on the path length of light scattered by aerosol particles and thus is related to the height of aerosol layers. The premise of this retrieval dated back to Hanel (1961) and Yamamoto and Wark (1961), who estimated cloud top pressure based on the amount of absorption by molecular $CO_2$ and $O_2$ above the cloud layer. More recently, a number of satellite sensors have been designed to capture

the $O_2$ absorption feature to provide cloud top height and ALH; among those are GOME, SCIAMACHY, MERIS, POLDER, DSCOVR, and most recently TROPOMI (Xu et al., 2018a and references therein).

Figure 2 illustrates the physical principle for sensing of ALH using the $O_2$ absorption spectroscopic approach, which relies on the fact that a scattering aerosol layer can scatter sun light back to space, shortening the path length of a photon traveling in the atmosphere, and reducing the chance of that photon being absorbed by $O_2$ molecules. As a result, an elevated

scattering layer enhances the Top Of the Atmosphere (TOA) reflectance within the $O_2$ absorption bands as detected by a satellite. In other words, for a given aerosol layer of fixed AOD placed at different altitudes, the higher the altitude, the larger the TOA reflectance, i.e., $I_H > I_L$ in Figure 2. This relationship contrasts to that of the reference continuum band, where the TOA reflectance is not sensitive to vertical location of the aerosol layer but depends only on the column AOD, i.e., $I_H = I_L$ for a given AOD. Based on this principle, the ratio of TOA reflectance between in-band and continuum band, or

the DOAS ratio, provides a practical way to infer the ALH (e.g., Dubuisson et al., 2009b; Duforêt et al., 2007; Xu et al., 2017).

To build the links between DOAS ratios and ALH, we simulate TOA reflectance as observed by EPIC measurements with the state-of-the-art Unified Linearized Vector Radiative Transfer Model (UNL-VRTM; described in section 3.5). Figure 3

shows simulations of the relationship between smoke ALH ($y$ axis) and DOAS ratios ($x$ axis) for EPIC channels in $O_2$ A and B bands and for various AOD values and surface reflectance ($A_s$, indicated by different colors). Those simulations were performed for a typical biomass-burning aerosol model as observed by EPIC for the geometry of $[\theta_0, \theta, \Delta\phi] = [42°, 37°, 165°]$, where $\theta_0$ and $\theta$ are solar and viewing zenith angles, and $\Delta\phi$ the relative azimuth angle between the sun and the viewer. As seen from Figure 3, the DOAS ratios in general increase with the rise of ALH. Meanwhile, the relationship is strongly dependent on the aerosol loading and surface reflectivity.

The sensitivity of DOAS to ALH is enhanced for lower surface reflectance and larger AOD. Conversely, the sensitivity decreases rapidly for smaller AOD (e.g., AOD = 0.1 in Figure 3a, d) or as $A_s$ increases, a condition where it is difficult to discriminate between the aerosol scattering contribution to TOA reflectance and contribution from the surface. Therefore, it is challenging to obtain ALH information for low aerosol loading conditions and over bright surfaces. This is consistent with findings of many previous information content studies for determining ALH from $O_2$ A and B band measurements (Ding et al., 2016; Dubuisson et al., 2009b). Duforêt et al. (2007) showed that the centroid altitude of a single aerosol layer over a dark surface can be retrieved from Polarization and Directionality of the Earth's Reflectances (POLDER) measurements with an error of less than 1 km when AOD is over 0.2. Similarly, Dubuisson et al. (2009b) suggested AOD > 0.3 and $A_s$ < 0.06 in order to achieve a retrieval accuracy of 0.5 km from POLDER.

With the above retrieval principles and challenges in mind, for our algorithm development we need to determine over what underlying Earth surfaces EPIC measurements have sufficient information to allow a practical retrieval of ALH. First and foremost, the surface reflectance must be low enough in one of the $O_2$ A and B bands to provide unambiguous signal from the aerosol. According to Figure 3, the DOAS-ALH sensitivity becomes substantial when $A_s$ < 0.1 in moderate aerosol loading conditions ($\tau_{680} = 0.4$). Hence, $A_s = 0.1$ is set as the upper threshold, above which an ALH retrieval will not be attempted, despite the sensitivity increases in high aerosol loading ($\tau_{680} = 1.0$) for higher $A_s$.

Figure 4 shows the magnitude of surface reflectance from the ASTER spectral library for typical Earth surface types (Baldridge et al., 2009). The water surface is dark, with reflectance lower than 0.03 in both the $O_2$ A and B bands. Over land, green vegetation is the only surface type that offers a reflectance below 0.1 in the $O_2$ B band ($A_s = 0.1$ is indicated by the red dotted line). In fact, $A_s$ in the $O_2$ A-band is considerably above 0.1 for all land types considered in Figure 4. These findings provide important guidance on the ALH algorithm design. Specifically, EPIC can retrieve ALH only over water and vegetated surfaces. In particular, ALH retrieval over vegetated surface would enlist DOAS ratios in the $O_2$ B-band. Additionally, retrieval of ALH requires a sufficiently high aerosol loading. We set AOD thresholds for retrieving ALH over both the water and vegetation per the algorithm overview to follow.

## 3. EPIC aerosol layer height retrieval algorithm

### 3.1 Algorithm overview

Figure 5 illustrates the processing of EPIC data and the ALH retrieval procedure. Briefly, the retrieval algorithm entails the following steps:

1. Calculate TOA reflectance in six EPIC visible and NIR bands (443, 551, 680, 688, 764, and 780 nm) from the calibrated EPIC level 1B digital data.

2. Identify EPIC pixels that are suitable for aerosol height retrieval. Through various tests, this step screens out pixels having clouds, over-water sun glints, and bright land surfaces, which are performed separately for water and land pixels. Surface pressure comes from MERRA-2 reanalysis data and we determine surface reflectance in EPIC bands using GOME-2 and MODIS surface products.

3. Aggregate the original EPIC pixels into a box of $3 \times 3$ individual pixels, an area with size of about 24 km at nadir. In many cases, not all pixels within a box are suitable for retrieval (i.e., cloud, glint, and bright land). If the number of available pixels within a box is not less than 4 (of the total of 9), calculate mean values of TOA reflectance, satellite geometries, and surface reflectance for the available pixels. Otherwise, do not conduct an aerosol retrieval for the box.

4. Invert the aggregated EPIC observations using pre-calculated lookup tables to obtain smoke ALH and AOD. The inversion uses a flexible spectral fitting strategy that considers the specific surface type.

While the retrieval procedure is based on our algorithm of retrieving dust ALH over ocean from EPIC measurements (Xu et al., 2017), it was upgraded in several ways. First, the algorithm is extended to retrieve ALH over vegetated land surface. The capability of inferring ALH over vegetation is predicated on the $O_2$ B band, where the surface reflectance is sufficiently low. Accordingly, the new algorithm uses methods separated for land and water surfaces in determining surface reflectance and screening cloud. Second, the algorithm incorporates a smoke aerosol optical model in order to retrieve biomass-burning smoke ALH. Third, rather than aggregating EPIC data from satellite pixels to regular latitude-longitude grids, the new algorithm retrieves over aggregated boxes, each consisting of an array of $3 \times 3$ EPIC pixels. This change is based on the consideration that geolocation of EPIC data has made significant improvements in the new version of level 1B data (Geogdzhayev and Marshak, 2018). Lastly, to obtain AOD and ALH the new algorithm employs a flexible spectral fitting strategy by considering the underlying surface reflectivity. In the following procedural subsections, we describe these changes in full detail.

## 3.2 Obtaining EPIC TOA reflectance

For this algorithm development, we used the EPIC Level 1B (L1B) Version 02 imagery data, available from NASA's Atmospheric Science Data Center (ASDC) at https://eosweb.larc.nasa.gov (last accessed Nov 26, 2018). Various pre- and post-launch calibrations were applied to the L1B EPIC data. EPIC visible and near-IR (NIR) channels at 443, 551, 680, and 780 nm were cross-calibrated with independent LEO satellite instruments, including the MODIS onboard the Terra and Aqua satellites (Geogdzhayev and Marshak, 2018). The two $O_2$ absorption channels (688 nm and 764 nm) were calibrated using lunar surface reflectivity with EPIC lunar observations at the time of full moon as seen from Earth (Ohtake et al.,

2010). The image data in EPIC L1B products is in digital units of counts/second, that is converted into reflectance for each visible and NIR channel using calibration factors provided at ASDC online: https://eosweb.larc.nasa.gov/project/dscovr/DSCOVR_EPIC_Calibration_Factors_V02.pdf (last accessed Nov 26, 2018). The TOA reflectance values are calculated as:

$$5 \quad R_\lambda = \frac{K(\lambda)C(\lambda)}{\mu_0}, \quad (1)$$

where $C(\lambda)$ is the EPIC measured signal in units of counts/second at the wavelength of $\lambda$, $K(\lambda)$ is a calibration factor, and $\mu_0$ is the cosine of solar zenith angle, $\theta_0$. Additionally, pixels that are far from nadir and strongly distorted by the Earth ellipsoid (e.g., view zenith angle $\theta > 70°$ or solar zenith angle $\theta_0 > 70°$) are excluded.

**3.3 Determining surface albedo and pressure**

Past studies show that an accurate ALH retrieval critically depends on the appropriate assumption of surface reflectance (Corradini and Cervino, 2006; Ding et al., 2016; Dubuisson et al., 2009a). As discussed in the preceding section, our retrievals are confined to water and dark vegetated surfaces. Land and water mask information for each EPIC pixel was determined using the GSHHG coastline database (Wessel and Smith, 1996). Following Xu et al. (2017), surface reflectance

over water surface was obtained from the GOME-2 surface Lambert-equivalent reflectivity (LER) database (Koelemeijer et al., 2003; Tilstra et al., 2017). GOME-2 LER products contain spectral LER albedo at 21 one-nanometer-wide channels, available globally for each month of the year at 0.25°×0.25° grid spacing (~25 × 25 km cells). From these data, we selected LER albedos at 440, 555, 670, 758, and 780 nm to represent surface reflectance for the nearest EPIC bands.

The Moderate-resolution Imaging Spectroradiometer (MODIS) BRDF/Albedo product (MCD43) provides parameters that

can be used to determine surface reflectance over land in the first seven MODIS channels (Lucht et al., 2000; Schaaf et al., 2002). Those parameters, including surface bidirectional reflectance distribution function (BRDF) and albedos, were inverted from the atmospherically-corrected (i.e., Rayleigh and aerosol components removed) MODIS reflectance observations from both the Terra and Aqua satellites over a 16-day period. We used black-sky albedo and white-sky albedo compiled in the Level-3 daily Climate Modeling Grid (CMG) Albedo Product (MCD43C3) at a spatial resolution of

0.1°×0.1°. Here, black-sky albedo is the directional hemispherical reflectance if the illumination comes only from the sun at solar noon. White-sky albedo is the bi-hemispherical reflectance under isotropic sky-light illumination. The actual bi-hemispherical reflectance can be computed from white-sky and black-sky albedos via a weighting coefficient that depends primarily on solar angle and columnar optical depth (Kokhanovsky et al., 2005; Lewis and Barnsley, 1994; Schaaf et al., 2002). Following Kokhanovsky et al. (2005), we calculated the Lambertian surface albedo in the 469, 555, 645, and 858 nm

MODIS bands by weighting the contributions of white-sky and black-sky albedos. It should be noted that the effect of non-Lambertian surface reflection may bias the ALH retrieval, because uncertainty in surface reflectance can substantially affect the ALH retrieval accuracy (see Appendix A). Nevertheless, this type of impact could be limited as EPIC's earth

observations are confined within an almost constant viewing geometry with scattering angles between 165° – 178°. Further studies are needed to examine the detailed impacts, which will be one of our future efforts.

The differences in spectral position and width of corresponding EPIC and MODIS channels may result in discrepancies. To compensate, we adjust MODIS reflectance values to equivalent EPIC bands. The adjustment factors, in the form of linear regression coefficients, were determined from analyzing USGS (United States Geological Survey) hyperspectral data for green vegetation. Figure 6a shows the selected reflectance spectra of 47 vegetation samples from the USGS Spectral Library Version 7 (Kokaly et al., 2017) and the spectral locations of EPIC (blue) and MODIS (red-dotted) bands. Linear regression of reflectance values in each EPIC band and corresponding MODIS band is illustrated as scatterplots in Figures 6c-h. The regression slope and offset, as well as coefficient of determination ($R^2$), are listed in Figure 6b. As shown, the reflectance between EPIC bands and their closest MODIS bands are highly correlated ($R^2$ from 0.96 to 1.0), and therefore, best-fitting equations in Figure 6b can be used to empirically derive EPIC surface reflectance from MODIS surface reflectance products.

The retrieval also requires auxiliary meteorological information to realistically characterize $O_2$ absorption properties. Surface pressure strongly affects the reflectance in $O_2$ absorption bands as observed from space, because it is a direct proxy for the columnar loading of air (and thus oxygen) molecules. Here, we enlisted surface pressure information from the Modern-Era Retrospective analysis for Research and Applications Version 2 (MERRA-2) datasets (Gelaro et al., 2017). MERRA-2's 1-hourly surface pressure at 0.5° by 0.675° grids were interpolated to the location and scan time of each EPIC pixel. In addition, the atmospheric temperature-pressure profile also impacts the width and strength of $O_2$ absorption lines. However, such influence on the radiative transfer is negligible for EPIC's 1-to-2-nm-wide bands. In this study, our algorithm employs a standard temperature-pressure profile representing the mid-latitude-summer atmosphere (McClatchey et al., 1972).

### 3.4 Masking clouds, sun glints, and bright land surface

After acquiring EPIC TOA reflectance and surface reflectance, the algorithm conducts a masking exercise to select clear-sky pixels suitable for aerosol retrieval. This includes the screenings of clouds, sun glint over water surfaces, and highly reflective land surfaces. Over both land and water, the cloud mask combines a set of brightness and homogeneity tests following Martins et al. (2002). The brightness tests screen out cloud pixels where EPIC TOA reflectance in two or three visible bands (443 and 680 nm over land; 443, 680, and 780 nm over water) exceed prescribed thresholds. The homogeneity test, on the other hand, identifies cloud pixels by evaluating the standard deviation of TOA reflectance at 443 and 551 nm within 3×3 neighboring pixels. Still, thin and small subpixel clouds can be missed in the relatively coarse resolution 8 km EPIC pixels, leading to an overestimation of AOD and an underestimation of ALH. Therefore, contamination by small-scale clouds is one of the known issues of retrieval quality and coupling higher resolution cloud mask information from

geostatistical sensors may help to overcome this issue. ~~Besides, cloud mask thresholds used in this work might need to be adjusted for operational applications.~~

In addition, EPIC pixels affected by sun glint and highly-reflective land surface are also removed from consideration by the mask. We identify the over-water glint area as those pixels having a glint angle smaller than 30° (Levy et al., 2013). Highly reflective land-surface pixels are identified using MODIS land surface products. Any EPIC pixel having NDVI below 0.2 or having a 680 nm surface reflectance larger than 0.1 is flagged as being a bright surface and excluded from the retrievals.

## 3.5 Look-up tables

Following Xu et al. (2017), the revised algorithm retrieves smoke ALH and AOD from EPIC measurements via a set of lookup tables (LUTs) constructed using the UNL-VRTM model. The UNL-VRTM (https:// unl-vrtm.org) is a radiative transfer testbed developed specifically for atmospheric remote sensing (Wang et al., 2014). By integrating the VLIDORT vector radiative transfer model (Spurr, 2006) and particulate scattering codes (Spurr et al., 2012), UNL-VRTM can perform simulations for two or more sets of aerosol microphysical properties, typically with aerosols in one fine mode and one coarse mode. It also incorporates HITRAN spectroscopic gaseous absorption with up to 22 trace gases (Rothman et al., 2013), allowing for accurate hyperspectral simulations of remote sensing observations (Xu et al., 2018b).

The LUTs for the current retrieval consist of simulated EPIC TOA reflectance at selected spectral bands for a set of AOD and ALH values under various atmospheric and observation scenarios (e.g., sun-earth-sensor geometry, surface reflectance, and surface pressure) as shown in Table 1. To build the LUTs, hyperspectral monochromatic radiances were simulated using UNL-VRTM and convolved with EPIC instrumental filter response functions for the selected six bands. Table 2 provides the spectral range, interval, and resolution (in terms of full-with at half maximum; FWHM) of the UNL-VRTM simulations.

The UNL-VRTM simulations assume a Gaussian-like aerosol profile characterized by a centroid altitude and a half-width parameter (Spurr and Christi, 2014; Xu et al., 2017). The centroid altitude is taken to represent ALH. A half-width of 1 km is assumed following Xu et al. (2017). This value was also used to derive AOD from UV observations by the Total Ozone Mapping Spectrometer (TOMS) and the Ozone Monitoring Instrument (OMI) (Torres et al., 1998). For this study, we implemented smoke optical properties calculated based on Lorenz-Mie scattering theory. Smoke refractive index was assumed to be 1.5 – 0.012i, following Dubovik et al. (2002), and its spectral dependence was neglected. We assumed smoke particles followed a bi-lognormal size distribution, as adopted for the MODIS dark-target aerosol algorithm (Levy et al., 2013; Remer et al., 2005). Specifically, fine-mode volume median radius and standard deviation were prescribed as $0.14 + 0.01\tau_{680}$ μm and 0.44, respectively. The coarse-mode counterparts are $0.14 + 0.01\tau_{680}$ μm and 0.80, respectively. The volume ratio between fine and coarse modes is $(0.01 + 0.3\tau_{680})/(0.01 + 0.09\tau_{680})$. Here, $\tau_{680}$ is the total AOD at 680 nm.

## 3.6 Strategy of flexible spectral fitting

The retrieval procedure involves two steps over both water and land surfaces and, at the same time, incorporates flexible spectral fitting that accounts for the specifics of surface reflectivity. First, the TOA reflectance in the EPIC "atmospheric

window" channels are matched with LUTs to determine AOD, since the TOA reflectance does not depend on ALH in these channels. Second, based on this estimated AOD, the DOAS ratios around the $O_2$ bands are fitted to estimate ALH. In each step, least-squared fitting is applied, and fitting residuals are reported.

In general, retrieving aerosol information from reflected solar radiation in vis-to-NIR wavelengths over land is more challenging than over ocean, since the satellite signal tends to be dominated by surface contributions over land. It is thus often the case that, for the same satellite instrument, separate over-land and over-ocean algorithms are developed with different strategies for characterizing surface reflectivity and band selection for fitting, e.g., MODIS aerosol algorithms (Remer et al., 2005). Similarly, our retrieval of smoke ALH from EPIC uses different band combinations over land versus over ocean, adjusted by surface type and spectral signature of smoke aerosol in the TOA reflectance.

In general, reflectance of water surfaces does not exceed 0.03 across the entire vis-to-NIR spectrum (Figure 4). Therefore, AOD can be determined by fitting TOA reflectance in all of the four "atmospheric window" channels at 443 nm, 551 nm, 680 nm, and 780 nm. In contrast, the 780 nm band is excluded for spectral fitting over vegetated land surface because of the high chlorophyll reflectance. Similarly, in the fitting of DOAS ratios, different weights are given for the ratios in the $O_2$ A and B bands for different surfaces, adjusted by the sensitivity of the DOAS ratio to ALH (e.g., Figure 3). The weighting coefficients are 0.4 for $R688/R680$ and 0.6 for $R764/R780$ over water surface, whereas these values are 0.9 and 0.1 over vegetated surface.

## 4. Retrieval demonstration

We apply our algorithm to six EPIC scenes over the Hudson Bay – Great Lakes area obtained during 25 – 26 August 2017, with three consecutive scenes considered on each day. We chose these cases primarily because smoke plumes prevailed over both water and vegetated surface, and there were available aerosol vertical profile measurements from CALIOP that could be used to validate the ALH retrievals. As shown in the EPIC RGB images (Figures 7a and 8a), smoke aerosols appeared as diffuse plumes emitted from wildfires in western Canada and transported over the western and southern portions of Hudson Bay (Peterson et al., 2018). The retrieved smoke ALH is shown in Figures 7b and 8b, and retrieved 680-nm AOD in Figures 7c and 8c. It is noted that data gaps in the AOD maps represent screened-out bright pixels due to either cloud or bright land surface; ALH retrievals were only available when AOD was larger than 0.2.

Obvious spatial variations are noted in retrieved smoke ALH and AOD. On August 25, smoke plumes had AOD values ranging from 0.1 to 0.45, with higher loading found at downwind regions in the south (Figure 7c). An ALH of 4 – 5 km was found over Hudson Bay, whereas the smoke altitudes decreased to 2 – 4 km over land off the bay's western and southern shores (Figure 7b). Southward, the ALH increased rapidly to 4 – 6 km towards the Great Lakes. By August 26, the smoke plumes had traveled southeast (Figure 8c). The smoke altitudes remained at 3 – 5 km over the eastern part of Hudson Bay, and 2 – 4 km over the bay's south side (Figure 8b). Altitudes of smoke plumes over the coast of northeastern U.S. were higher than 5 km. Aside from spatial variations, the retrievals also revealed diurnal changes of ALH and evolution of the

smoke plumes. For instance, the ALH of smoke plumes over the Hudson Bay and the north side of the Great Lakes rose by about half km within two hours from local morning to afternoon on August 25 (Figure 7b).

## 5. Retrieval validation

For the validation of the EPIC-retrieved ALH, we used CALIOP Level 2 version 4.10 aerosol extinction profiles at 5 km spatial resolution, retrieved from CALIOP observations of attenuated backscatter at 532 nm (Young and Vaughan, 2009). To facilitate quantitative comparison of aerosol altitude, we used a mean extinction height calculated from the CALIOP extinction profile, following Koffi et al. (2012):

$$\text{ALH}_{\text{CALIOP}} = \frac{\sum_{i=1}^{n} \beta_{\text{ext},i} \, Z_i}{\sum_{i=1}^{n} \beta_{\text{ext},i}} \quad (2)$$

Here, $\beta_{\text{ext},i}$ is the aerosol extinction coefficient (km$^{-1}$) at 532 nm at level $i$ and $Z_i$ is the altitude (km) of level $i$. Thus, ALH$_{\text{CALIOP}}$ represents an effective ALH weighted by aerosol extinction signal at each level and is consistent with ALH as defined in our EPIC algorithm.

In the CALIOP Level 2 products, aerosol extinction is only retrieved for the layers where aerosols are detected, as permitted by the instrument's signal-to-noise ratio (SNR). Atmospheric layers with no aerosols detected are classified as "clear air" and assigned an aerosol extinction coefficient of 0 km$^{-1}$. The detection limits are defined in terms of backscattering ratio, which depends on an aerosol lidar ratio (Vaughan et al., 2009). As such, aerosol layers that span a large altitude range frequently remain undetected (Toth et al., 2018), particularly for absorbing aerosols in the day time. As indicated by Winker et al. (2013), aerosol extinction threshold in a daytime CALIOP scan is $0.01 - 0.03$ km$^{-1}$ for 80-km horizontal averaging resolution and increases to 0.07 km$^{-1}$ for 5-km horizontal averaging resolution. In reality, aerosols are ubiquitous throughout the troposphere, though the concentration can be very low in the free troposphere. However, excluding the "clear-air" layers would lead to a biased estimate of ALH$_{\text{CALIOP}}$, particularly for cases of predominantly "clear-air" layers present below an elevated aerosol layer. To compensate for this bias, we use an exponentially-decayed background aerosol extinction profile to provide a proxy for aerosol extinction coefficients of these undetected aerosol layers within the troposphere. The background profile has a columnar AOD of about 0.07 at 532 nm, according to Tomasi and Petkov (2014), who use various lidar measurements to characterize the summertime Arctic atmosphere. Though aerosol extinction within the undetected aerosol layers by no means follows the background extinction profile identically, adding such a background aerosol extinction profile could, to the first order, correct the potential bias in ALH$_{\text{CALIOP}}$. Therefore, we consider ALH$_{\text{CALIOP}}$ with and without the imposed background aerosol in the following validation analysis.

Figure 9 presents the evaluation of our ALH retrieval against CALIOP observations, showing that the ALH retrieval captures the overall spatial variability of ALH as seen by CALIOP. Quantitatively, 67% and 59% of the retrievals are within 0.5 km difference from the counterparts of ALH$_{\text{CALIOP}}$ on August 25 and 26, respectively. Considering all EPIC-CALIOP ALH pairs, over 65% of ALH retrievals are within an uncertainty envelope of 0.5 km (Figure 10a-b), and EPIC ALH has a mean

bias of –0.13 km and a root-mean-squared-error (rmse) of 0.58 km, capturing 52% variation of the $ALH_{CALIOP}$ (Figure 10a). If background aerosol is imposed in the CALIOP ALH calculation, our retrieved EPIC ALH is found to have a bias of 0.23 km and a rmse of 0.57 km (Figure 10b).

The EPIC 680 nm AOD retrievals during the two days were compared against 675 nm AOD observations at nine AERONET sites (Table 3). The collocation method follows Ichoku et al. (2002), but was modified to compare EPIC AOD retrievals over $3 \times 3$ pixels centered at the AERONET sites with 1-hour AERONET AOD observations around the EPIC scan time. Collocated AERONET AOD values at each site are shown as circles in Figures 7c and 8c. A comparison of EPIC and AERONET AODs is shown in Figure 10c. The collocated AOD pairs, though with limited data samplings, have over 77% falling in an uncertainty envelope of $\pm (0.05 + 0.1AOD)$ with a coefficient of determination ($R^2$) of 0.54. The EPIC AOD shows a positive bias of 0.03 and a rmse of 0.05, which was dominated for four subsets of EPIC AODs and was likely caused by cloud contamination.

## 6. Implication to the retrieval of UV absorbing aerosol properties

One important implication of this study is that the retrieved ALH can provide complementary height information for determining absorbing aerosol properties from EPIC's UV bands. EPIC also measures backscattered UV radiances at 340 nm and 388 nm, which was designed to detect and retrieve optical properties of UV-absorbing aerosols like mineral dust and smoke by using the UV aerosol index (UVAI) (Marshak et al., 2018). UVAI quantifies the difference in spectral dependence between measured and calculated near UV radiances assuming a purely molecular atmosphere (Torres et al., 1998). Physically, inferring aerosol properties from those UV bands requires the characterization of aerosol height, since UV radiance is sensitive to aerosol vertical distribution. For example, Jeong and Hsu (2008) retrieve aerosol single scattering albedo (SSA) from OMI radiance via synergic use of AOD from MODIS and ALH from CALIOP. Currently, EPIC's UV aerosol algorithm utilizes climatological aerosol height data set derived from CALIOP observations (Torres et al., 2013). However, these static climatological data may fail to capture the dynamic variation of aerosol height, and thus induce uncertainties in the UV-retrieved aerosol properties. With the aerosol height and loading available from the EPIC's $O_2$ A and B bands, these closures are now possible with measurements from a single instrument. Such synergy can be also applied to the TROPOspheric Monitoring Instrument (TROPOMI) and the Plankton, Aerosol, Cloud, ocean Ecosystem (PACE) satellite, both of which obtain hyperspectral measurements from the UV to the NIR covering the $O_2$ A and B bands (Omar et al., 2018; Veefkind et al., 2012).

Here, we compare our retrieved ALHs with EPIC UVAI to illustrate the importance, as well the potential benefit, of the ALH retrievals to EPIC's UV aerosol products. The EPIC UVAI data are publicly available at https://eosweb.larc.nasa.gov/project/dscovr/ (last accessed Nov 26, 2018). Figure 11 shows the UVAI of the same EPIC scenes on August 25. These smoke plumes are highlighted by large values of UVAI, which are in contrast to clouds that typically exhibit a UVAI close to zero (Torres et al., 1998). Since UVAI is a function of ALH, AOD, and SSA, its correlation with ALH varies with AOD. As shown in Figure 12, the sensitivity of UVAI to ALH, as well as the correlation

between them, increases as AOD increases. In particular, high AOD values (e.g., over 0.4) may result in UVAI values from less than 1 to about 4, depending on the ALH (Figure 12, bottom-right panel). Therefore, the use of ALH derived from the EPIC $O_2$ bands is expected to improve EPIC UV aerosol retrievals.

## 7. Summary and discussions

We extend our retrieval algorithm for retrieving over-water dust ALH from EPIC measurements (Xu et al., 2017) to biomass burning smoke aerosols over both water and vegetated land surfaces. The new algorithm uses earth-reflected radiances in EPIC's visible and NIR bands and incorporates flexible spectral fitting that accounts for the specifics of vegetation and water surface reflectivity. The fitting procedure first determines AOD using four EPIC atmospheric window bands (443 nm, 551 nm, 680 nm, and 780 nm), and then uses the DOAS ratios formulated in the $O_2$ A and B bands (688 nm and 764 nm) to

derive the ALH that represents an optical centroid altitude. ALH retrieval over vegetated surface primarily takes advantage of the measurements in the $O_2$ B band, where surface reflectance is sufficiently low to yield aerosol height information. Surface reflectance values are specified using MODIS surface products over land and GOME-2 LER products over ocean.

We applied the algorithm to six EPIC images, with three images on each day, having biomass burning plumes over the United States and Canada acquired on 25 – 26 August 2017. The algorithm is able to retrieve AOD and ALH multiple times

15 daily over both water and vegetated land surfaces. The retrieved ALHs were validated against CALIOP extinction weighted aerosol height ($ALH_{CALIOP}$), showing EPIC retrieved ALH has a rmse of 0.58 km and captures 52% variation of the $ALH_{CALIOP}$, and 65% of EPIC and CALIOP ALH pairs are within an uncertainty envelope of 0.5 km. EPIC retrieved AODs are validated against AERONET AOD observed at nine sites, indicating a rmse of 0.05 and over 77% of EPIC AOD retrievals fall within the error envelope of $\pm (0.05 + 0.1AOD)$. In addition, by comparing the retrieved ALH and EPIC

UVAI, we show that the aerosol height information retrieved by the present algorithm can potentially benefit the retrieval of aerosol properties from the EPIC UV bands.

The over three years of data recorded thus far by EPIC (July 2015 – present) offers an opportunity to characterize aerosol height from diurnal to seasonal scales. Such datasets with well-characterized uncertainties are valuable for evaluating aerosol vertical distribution simulated by climate models. A follow-on study, based on the current work and Xu et al. (2017), will

examine the long-term global retrieval of ALH for dust and smoke aerosols. Challenges for such a global retrieval include the heterogeneity of aerosol species, whereas our efforts have been mainly focused on single-species (dust or smoke) aerosol assumptions. In fact, mixtures of aerosol are more commonly found in nature. Furthermore, the currently available EPIC UVAI products can help identify elevated absorbing aerosols (mainly, dust and smoke), which can then be used to determine the retrieval targets on any EPIC image.

**Appendix A. Sensitivity and Error Analysis**

In this appendix, we investigate the sensitivity of EPIC measurements in the $O_2$ A and B bands to aerosol vertical profile shape, aerosol optical properties, and surface reflectance assumed in the retrieval algorithm. Then, we estimate potential

ALH retrieval errors due to the uncertainties in these parameters. For this case, ALH is derived from two DOAS ratios ($\rho$) in the O₂ A and B bands. Mathematically, the retrieval error ($\hat{\epsilon}$) of ALH can be estimated using the optimal estimation approach (Xu and Wang, 2015):

$$\hat{\epsilon}^{-2} = \mathbf{K}^T \mathbf{S}_\epsilon^{-1} \mathbf{K}, \qquad (A.1)$$

where $\mathbf{K}$ is the Jacobian matrix of $\rho$ with respect to ALH, and $\mathbf{S}_\epsilon$ is the observation error covariance matrix for $\rho$. In this study, we assume no error correlation between the two DOAS ratios; That is, $\mathbf{S}_\epsilon$ is a diagonal matrix with its elements equal to the error variance for $\rho$. Observation error variance matrix $\mathbf{S}_\epsilon$ can be characterized as a sum of two terms

$$\mathbf{S}_\epsilon = \mathbf{S}_\rho + \mathbf{S}_m. \qquad (A.2)$$

Here, $\mathbf{S}_\rho$ is the error covariance matrix describing the uncertainty for EPIC measurements. $\mathbf{S}_m$ is the covariance matrix for forward model errors caused by inaccurate model assumptions and uncertainties in model parameterizations, and $\mathbf{S}_m$ can be calculated by

$$\mathbf{S}_m = \mathbf{K}_b^T \mathbf{S}_b \mathbf{K}_b, \qquad (A.3)$$

where $\mathbf{S}_b$ is the error covariance matrix describing uncertainties of the vector of model parameters $\mathbf{b}$, and $\mathbf{K}_b$ is Jacobian matrix of $\rho$ with respect to $\mathbf{b}$. In the following analysis, we consider four parameters for the vector $\mathbf{b}$, i.e., the half-width parameter ($\sigma_H$) defining the quasi-Gaussian aerosol vertical profile, the 680-nm AOD ($\tau_{680}$), aerosol single scattering albedo (SSA), and surface reflectance $A_s$.

Figure A1 presents the UNL-VRTM simulated DOAS ratios $\rho$ in EPIC O₂ A and B band (a), and their Jacobian gradients with respect to ALH (b) and with respect to the above-mentioned four model parameters (c-f) for the vector $\mathbf{b}$ for various surface and aerosol loading conditions. Figure A2 shows the estimated ALH retrieval errors owing to EPIC observation error and uncertainties in the assumed model parameters for water (with $A_s = 0.05$ in both O₂ A and B bands) and vegetation (with $A_s = 0.30$ in A band and 0.05 in B band) surfaces. The findings from these Jacobian and error analysis results can be summarized as:

- The sensitivity of $\rho$ with respect to ALH (Figure A1b) is weak for aerosols confined in the lower atmosphere (ALH below 1.5 km). The sensitivity increases rapidly with the increase of ALH, peaks for ALH about 4 km, and then deceases slightly for higher elevated aerosols. The sensitivity in the O₂ A band is stronger than that in the O₂ B band, and it is stronger for low surface reflectance and high aerosol loading. By considering 2% uncertainty for the EPIC DOAS ratios (Geogdzhayev and Marshak, 2018), the ALH retrieval error ($\hat{\epsilon}_0$) is less than 0.5 km for the water surface and is less than 0.75 km for the vegetated surface if ALH is over 1.5 km (black curves in Figure A2a-b). This error characterization is consistent with the retrieval error (0.57 – 0.58 km) as estimated through the validation with CALIOP data.

- The DOAS ratios $\rho$ have a negative sensitivity to $\sigma_H$ for elevated aerosol, and the sensitivity turns to positive for ALH below 1.5 km (Figure A1c). An error of 0.5 km $\sigma_H$ may cause up to 0.3 km retrieval error for ALH (blue curves in Figure A2c-d).

- The DOAS ratios $\rho$ have a positive sensitivity to $\tau_{680}$, and the sensitivity increases as ALH increases (Figure A1d). An error of 0.05 in $\tau_{680}$ can lead to a retrieval error from 0 to 0.6 km for ALH, depending on the aerosol loading and surface type (orange curves in Figure B1c-d).

- DAOS ratios are sensitive to SSA to some degree, especially for large AOD values (Figure A1e). However, the sensitivity to SSA is much less overwhelmed than the sensitivity to AOD and surface reflectance because the reflectance at TOA depends more on surface reflectance and AOD (than SSA in relative sense).. As a result, SSA only has marginal impact to the ALH retrieval error (green curves in Figure A2c-d), which is consistent with findings by Sanders et al. (2015).

- The DOAS ratio $\rho$ has a negative sensitivity to $A_s$, and the sensitivity increases as ALH increases (Figure A1f). An error of 0.01 sensitivity to $A_s$ can result in a retrieval error of 0.1 – 0.6 km for ALH over water surface and 0.1 – 0.4 km over vegetation surface, depending on the aerosol loading (purple curves in Figure B1c-d).

- If including possible uncertainties from these four parameters in vector **b** and EPIC measurement error, the estimated ALH retrieval errors ($\hat{\epsilon}_{All}$) are shown as red curves in Figure B2a-b. For elevated smoke aerosols with $\tau_{680} = 0.4$, $\hat{\epsilon}_{All}$ ranges from 0.4 km t0 1.0 km for water surface and from 0.7 km to 1.25km for vegetation surface; the $\hat{\epsilon}_{All}$ range for $\tau_{680} = 1.0$ is reduced to 0.3 – 0.5 km for water 0.4 – 0.6 km for vegetation surface.

**Acknowledgements**

This research is in part supported by the NASA DSCOVR Earth Science Algorithms Program (Grant No. NNX17AB05G managed by Richard S. Eckman) and in part supported by the Office of Naval Research (ONR) Multidisciplinary University Research Initiatives (MURI) Program under award No. N00014-16-1-2040. We acknowledge the computational support from the High-Performance Computing group at The University of Iowa and the UMBC High Performance Computing Facility (HPCF). The data presented in this paper can be obtained through email to the corresponding authors (X. Xu and J. Wang) of this paper. EPIC Level 1B (L1B) Version 02 imagery data is obtained from the NASA Atmospheric Science Data Center (ASDC). We also acknowledge the AERONET program for providing AOD data retrieved from sunphotometers. We thank Dr. Feng Xu and two other anonymous reviewers for their insightful comments to the paper.

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

**Tables**

Table 1. Adopted parameters for generating LUTs.

| Parameter[*] | Prescribed values for LUTs |
|---|---|
| AOD at 680 nm | 0.0, 0.1, 0.2, 0.4, 0.7, 1.0, 1.5, 2.0, and 3.0 |
| ALH above surface (km) | 0 to 15 km at 1 km interval |
| Surface reflectance | 0, 0.05, 0.1, 0.2, 0.3, 0.4, and 0.6 |
| $\theta_0$ and $\theta$ (°) | 0° to 72° at interval of 6°, and $\lvert\theta_0 - \theta\rvert < 15°$ |
| $\Delta\varphi$ (°) | 0° to 180° at interval of 12°, |
| Surface pressure (hPa) | 700, 800, 900, and 1050 |

[*]$\theta_0$ indicates the solar zenith angle, $\theta$ the satellite viewing zenith angle, and $\Delta\varphi$ the relative azimuth angle

Table 2. Spectral settings of UNL-VRTM for constructing the LUTs

| EPIC channel (nm) | Spectral range (nm) | Spectral interval (nm) | FWHM (nm) |
|---|---|---|---|
| 443 | 440 – 445 | 0.1 | 0.2 |
| 551 | 549 – 555 | 0.1 | 0.2 |
| 680 | 675 – 685 | 0.1 | 0.2 |
| 688 | 685 – 690 | 0.01 | 0.02 |
| 764 | 760 – 766 | 0.01 | 0.02 |
| 780 | 776 – 782 | 0.1 | 0.2 |

Table 3. AERONET sites selected for AOD validation

| Site name | Latitude (°) | Longitude (°) | $N_{\mathrm{AOD}}$* |
|---|---|---|---|
| Billerica | 42.53 | − 71.27 | 3 |
| Churchill | 58.74 | − 93.82 | 3 |
| Egbert | 44.23 | − 78.78 | 2 |
| Lake_Erie | 41.83 | − 83.19 | 2 |
| LISCO | 40.76 | − 73.34 | 4 |
| NEON_UNDE | 46.23 | − 89.54 | 2 |
| Pickle_Lake | 51.45 | − 90.22 | 1 |
| Thompson_Farm | 43.11 | − 70.95 | 3 |
| Toronto | 43.79 | − 79.47 | 2 |

* $N_{\mathrm{AOD}}$ is the number of collocated AERONET AOD values

**Figures**

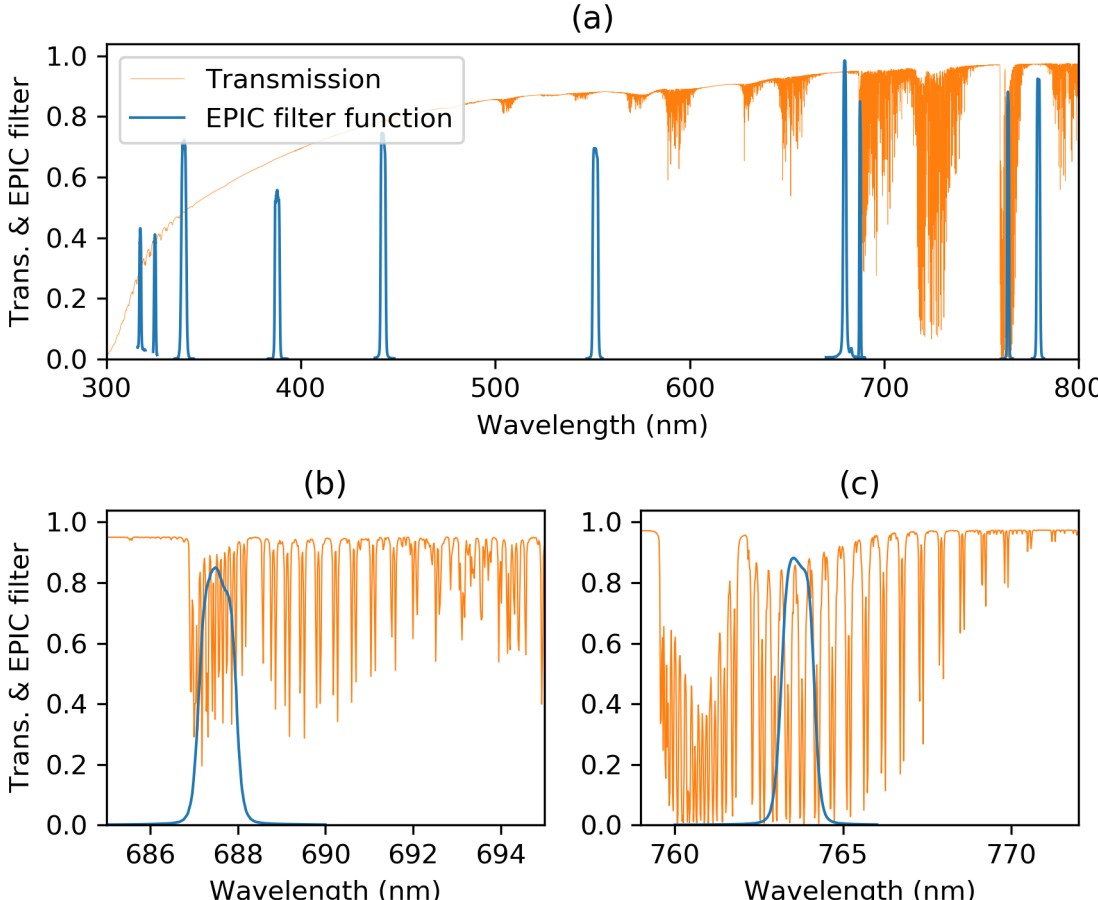

5   **Figure 1.** EPIC instrument filter response function (blue) and atmospheric spectral transmission (orange). Panel (a) includes all ten EPIC bands, whereas panels (b) and (c) show zoom-ins for the 688-nm channel in the $O_2$ B band and the 764-nm channel in the $O_2$ A band, respectively. Here the atmospheric transmission is simulated by the UNL-VRTM model, with a spectral step size and a spectral full width at half maximum of 0.02 nm.

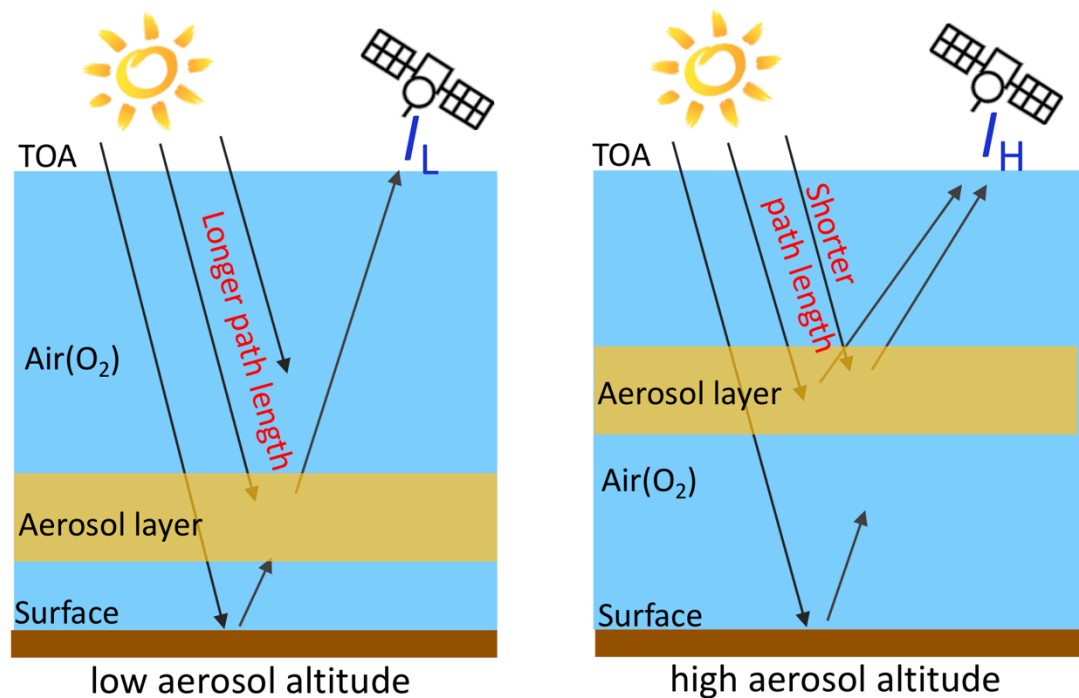

5 **Figure 2.** Schematic diagram of the physical principle of remote sensing of aerosol layer height in the $O_2$ absorption bands. Shown is the same aerosol layer located at two different altitudes in the atmosphere. Due to the scattering by aerosol particles, photons scattered by a higher aerosol layer (right panel) would travel a shorter pathlength to reach the satellite sensor than those scattered by the lower-altitude aerosol (left panel), resulting in less absorption by $O_2$ and thus a higher radiance value detected by the satellite (i.e., $I_H > I_L$).

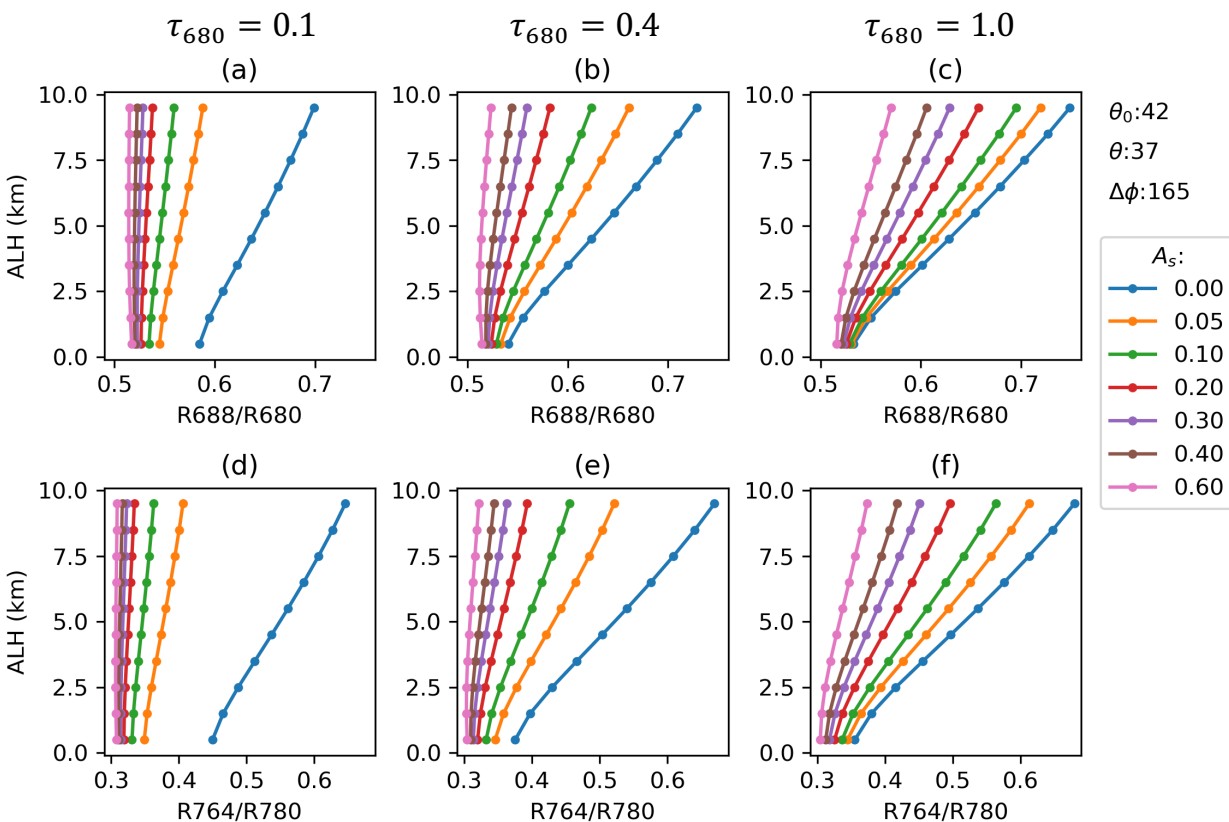

**Figure 3.** Sensitivity of DOAS ratios of TOA reflectance around the O₂ A and B bands to smoke ALH as simulated by UNL-VRTM for different AOD and surface reflectance ($A_s$) values. Simulations were performed for a typical biomass-burning aerosol model as observed by EPIC for the geometry of $[\theta_0, \theta, \Delta\phi] = [42°, 37°, 165°]$, where $\theta_0$ and $\theta$ are solar and viewing zenith angles, and $\Delta\phi$ the relative azimuth angle.

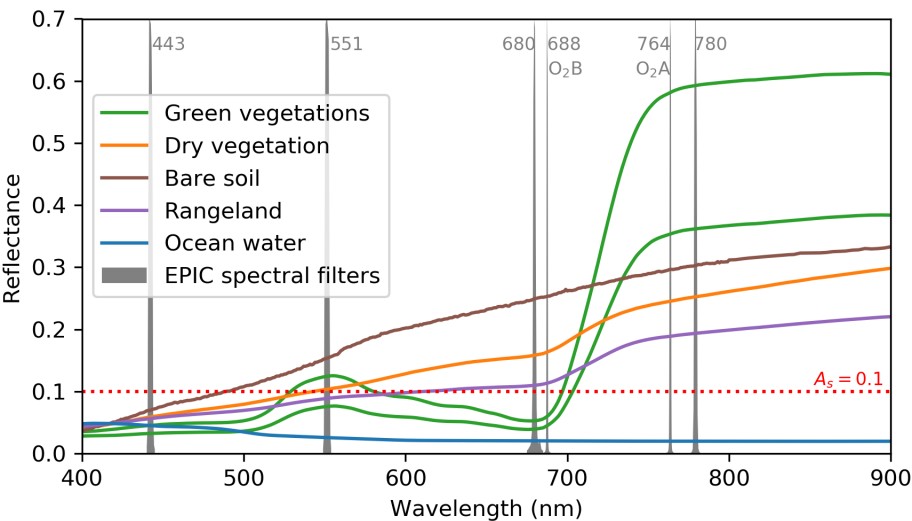

**Figure 4.** Surface reflectance spectra (from the ASTER spectral library; Baldridge et al., 2009) for various surface types in the visible-to-NIR range, with selected EPIC spectral bands within this range shown in gray.

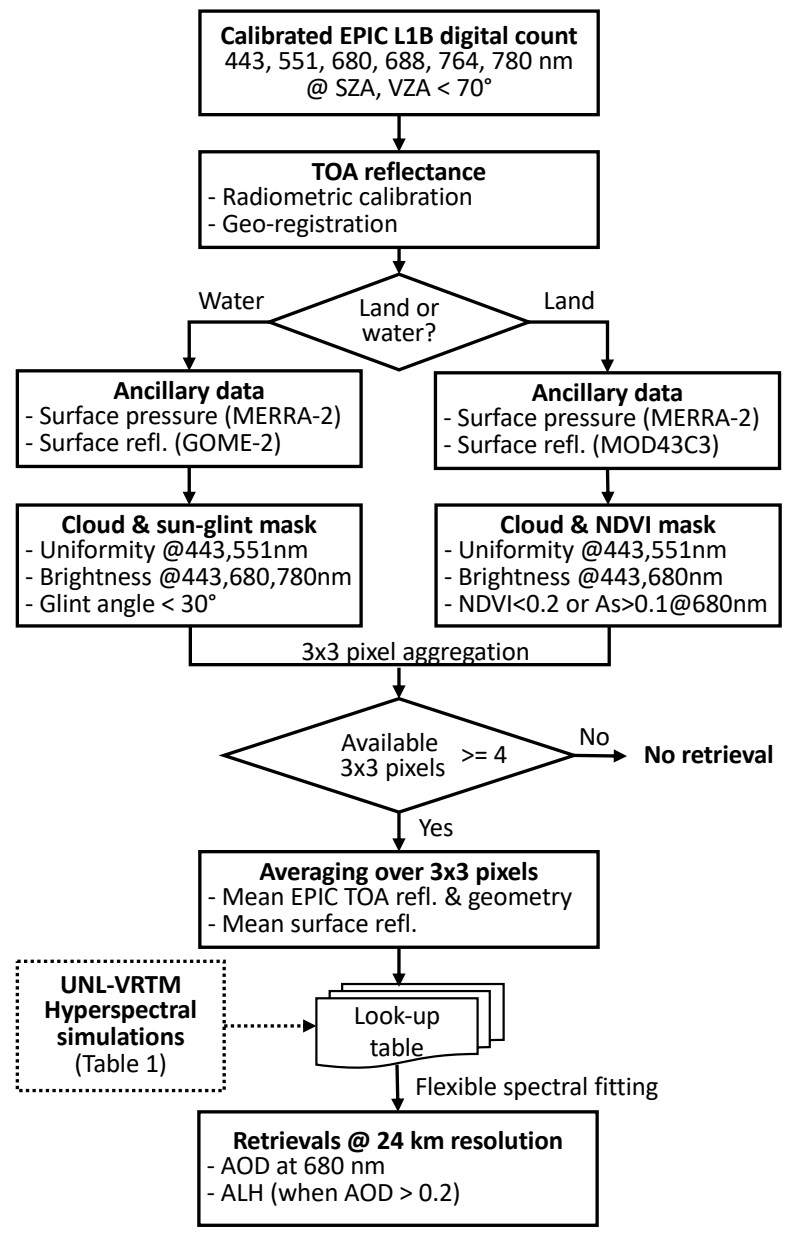

**Figure 5.** Flowchart of the AOD/ALH retrieval algorithm.

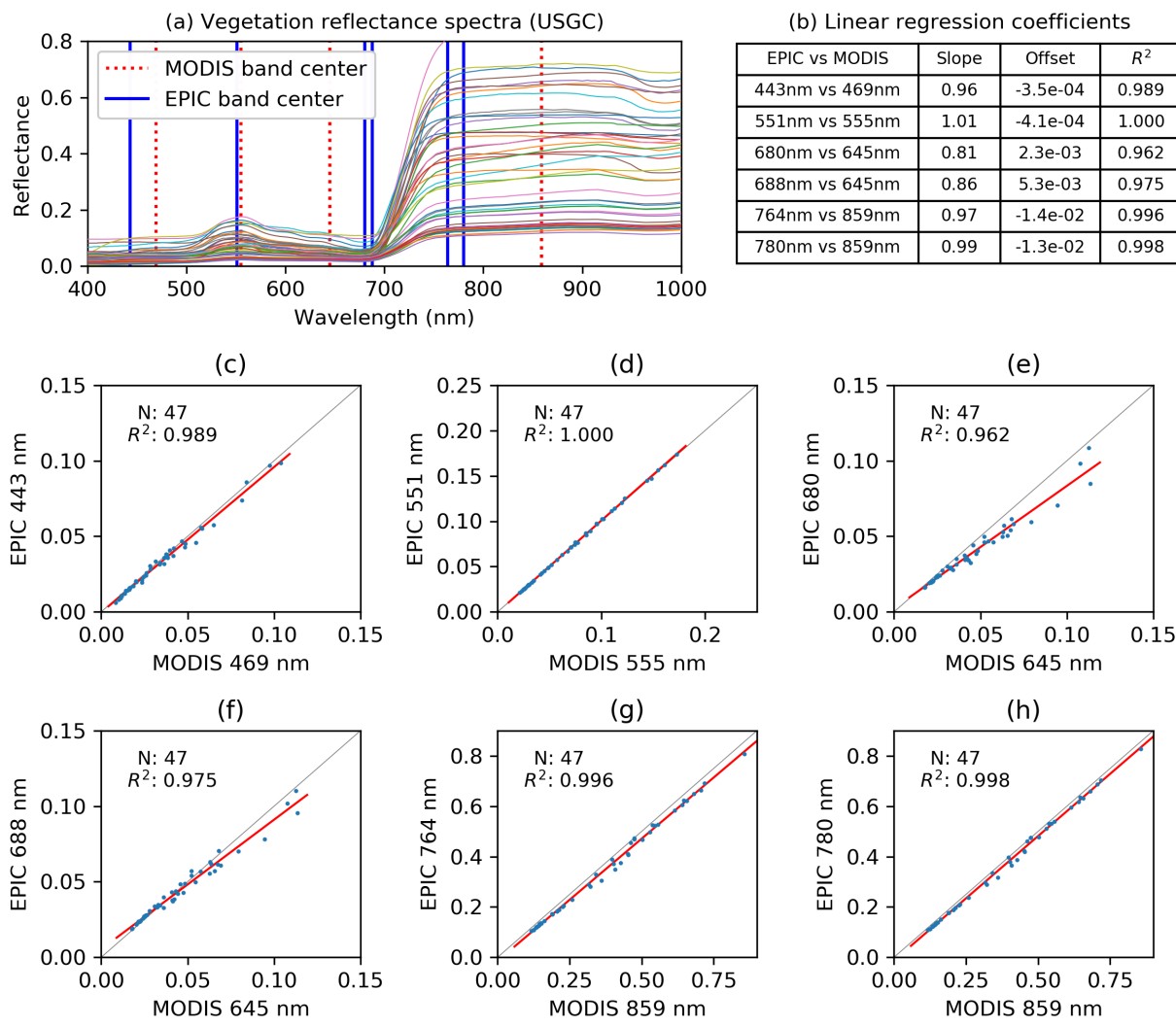

**Figure 6.** Determination of surface reflectance from MODIS surface albedo products. (a) USGS reflectance spectra for different vegetation samples, and (b) statistics for regression (i.e., slope, offset, and coefficient of determination, $R^2$) of reflectance between different EPIC bands, marked as red dotted lines in (a), and the respective closest MODIS bands, marked as blue lines in (a), according to the spectral library. (c-f) Scatterplots of reflectance in each EPIC band versus reflectance in the corresponding MODIS bands. Linear regression lines are shown in red.

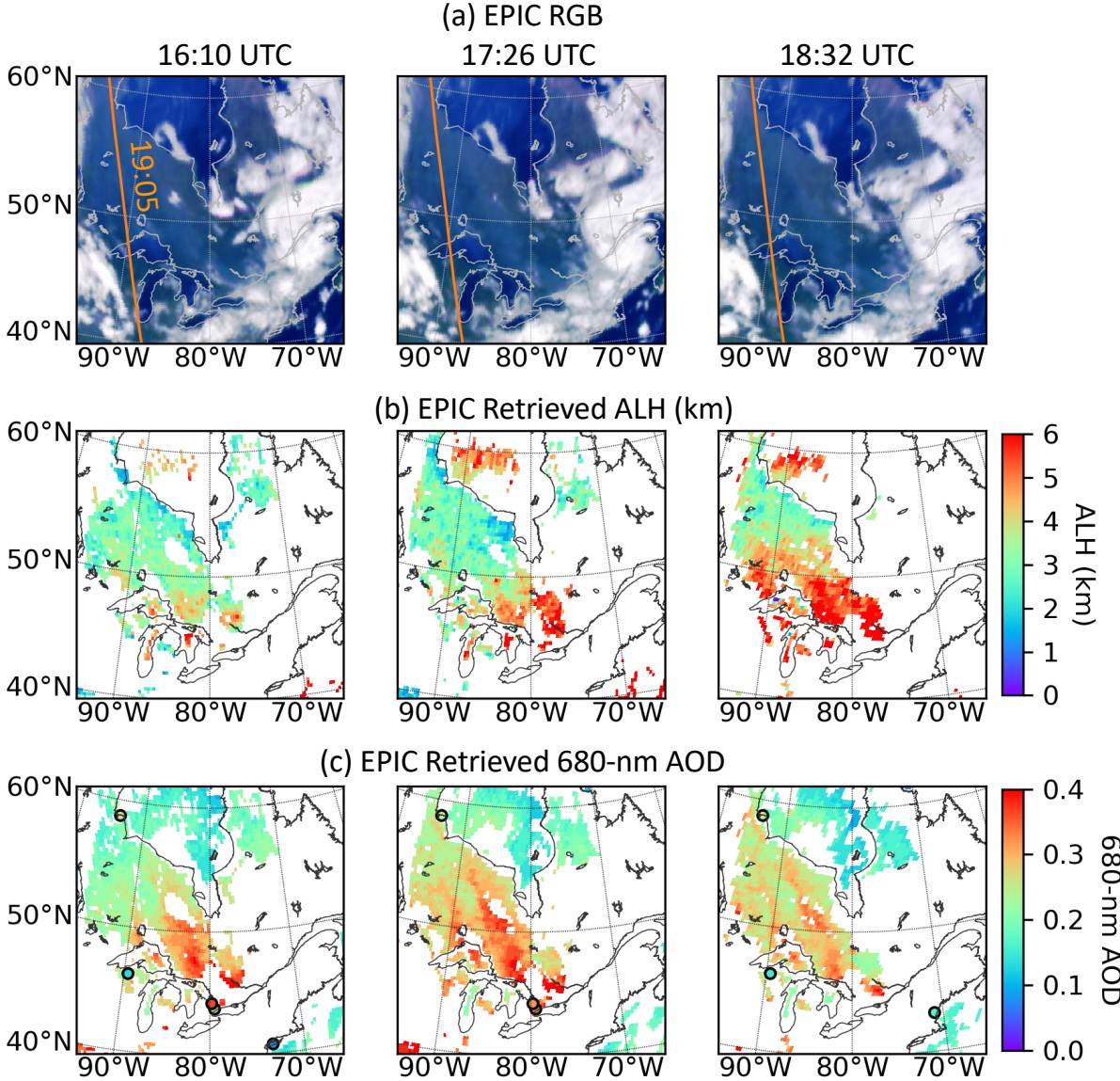

**Figure 7.** Demonstration of ALH and AOD retrievals for three EPIC scenes acquired on 25 August 2017 around UTC times of 16:10, 17:26, and 18:32. (a) RGB composites of EPIC 443 nm, 551 nm, and 680 nm data. The gold line indicates the CALIOP sub-orbital track with an overpass time of 19:05 UTC. (b) Retrieved smoke ALH when 680-nm AOD is larger than 0.2. (c) Retrieved 680 nm AOD. Small circles on AOD maps represent AOD values observed at corresponding AERONET sites within two hours of the EPIC scan time.

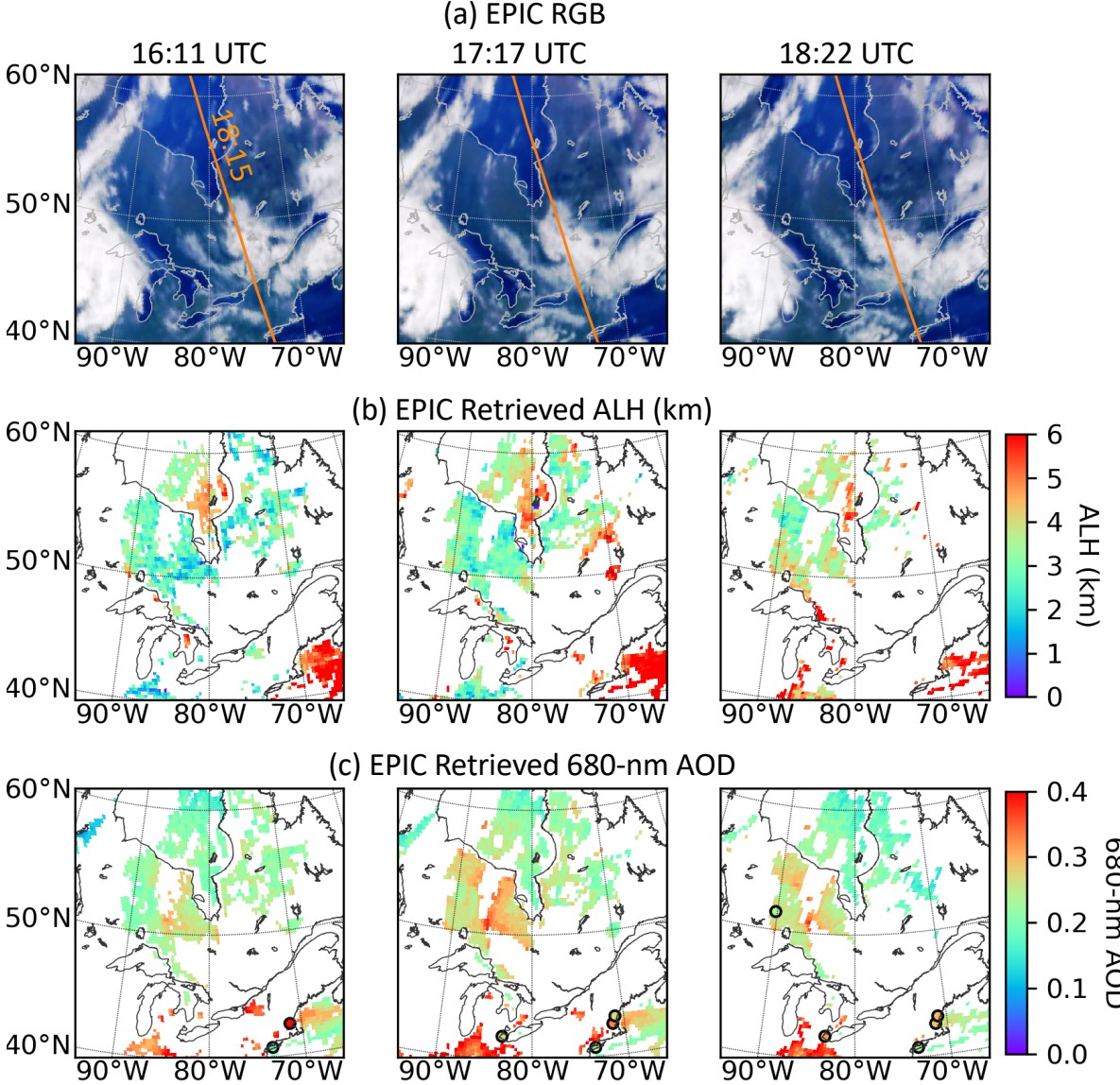

**Figure 8.** Same as Figure 7 but for three EPIC scenes captured on 26 August 2017 at UTC times of 16:11, 17:17, and 18:22. The CALIOP overpass was at 18:15 UTC.

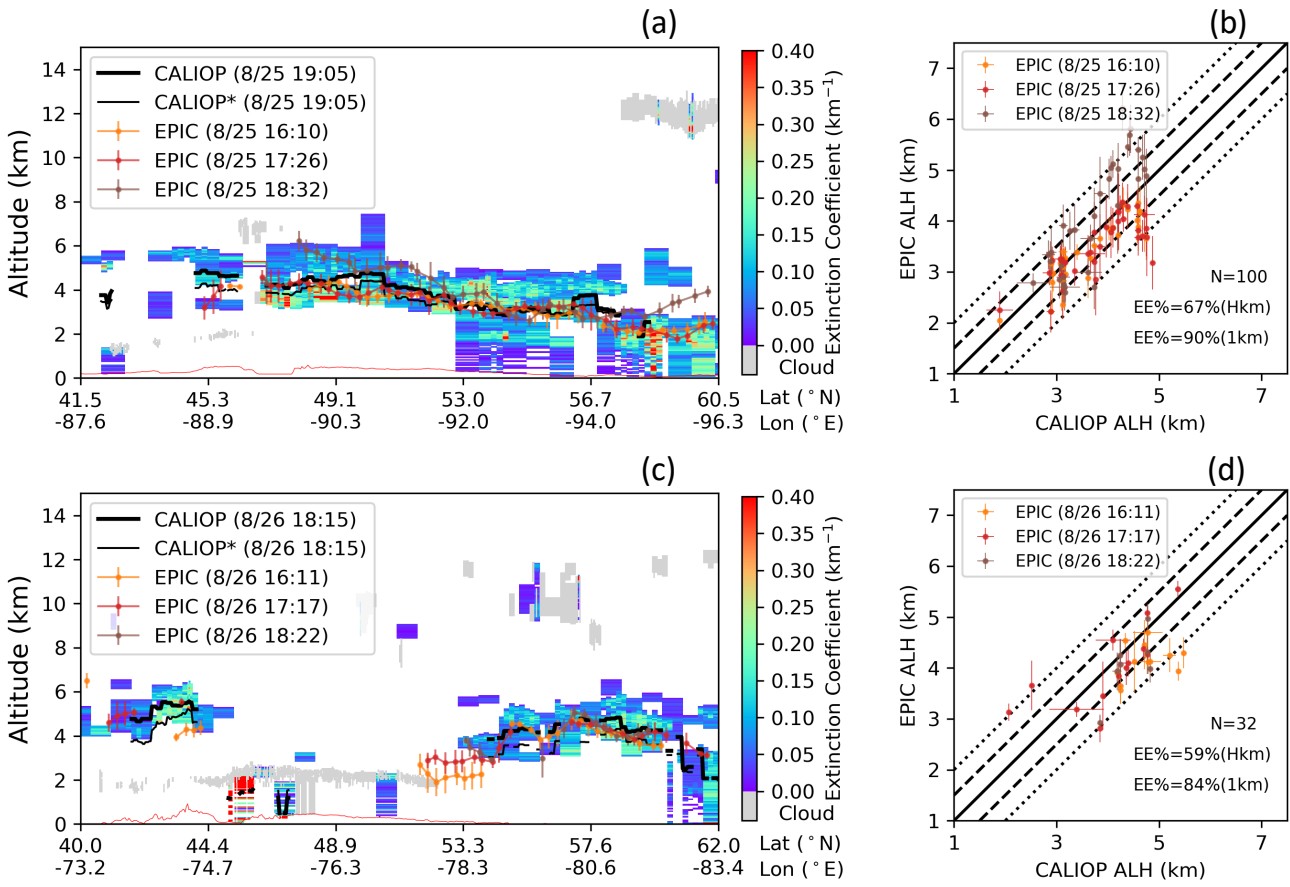

**Figure 9.** Comparison of ALH retrieved from EPIC and CALIOP level-2 aerosol extinction profile in both 'curtain' view (left column) and 'scatterplot' view (right column) for two-day overpasses. The CALIOP orbital tracks are marked on EPIC RGB images in Figures 7–8. Different EPIC scan times are marked with different line/symbol colors in the comparison. The error bar for EPIC ALH represents the standard deviation for an array of 3x3 24-km retrieval pixels, while that for CALIOP ALH represents the standard deviation of over 5 adjacent CALIOP 5-km footprints. Also shown in the left panels is the CALIOP* ALH (thin black curve) calculated with a background aerosol profile imposed for undetected aerosol layers (see text for detail), whereas bold black curve represents the CALIOP ALH without considering background aerosol.

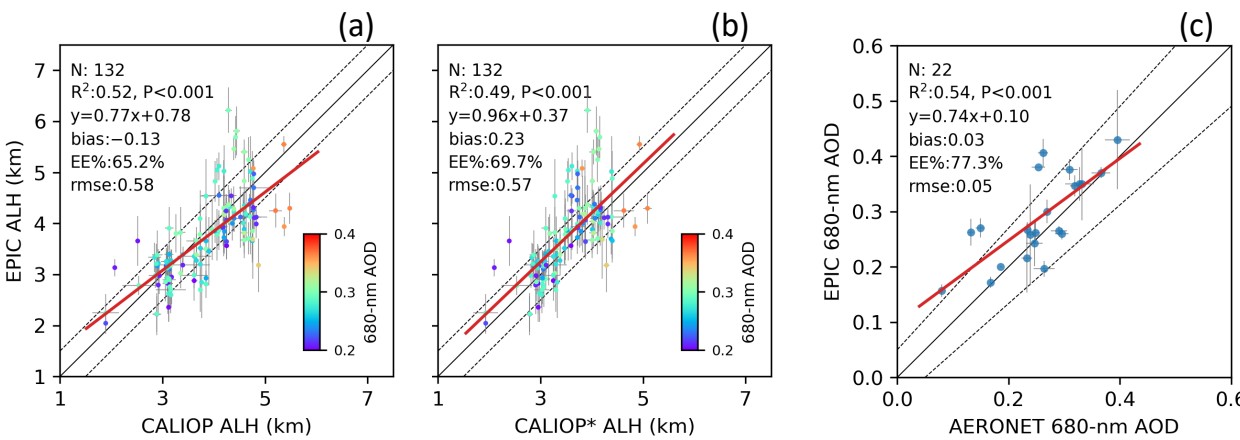

**Figure 10.** Comparison of EPIC ALH and AOD with the corresponding CALIOP and AERONET measurements. (a) Scatterplot of EPIC ALH versus CALIOP ALH by including all EPIC-CALIOP pairs shown in Figure 9. The color of each scatter point represents the EPIC 680-nm AOD value for the same EPIC pixels. (b) Same as panel (a) but for CALIOP ALH calculated with a background aerosol profile imposed for undetected aerosol layers. (c) Scatterplot of EPIC 680-nm AOD versus AERONET 675-nm AOD, collocated at 9 AERONET sites listed in Table 3. The dotted lines in both the scatterplots represent error envelops, i.e., ± 0.5 km for ALH and ± (0.05 + 10%) for AOD. Also annotated are the one-to-one line (solid black line), linear regression fit (red line), number of scatters points (N), coefficient of determination ($R^2$), significance level (P), linear regression equation, bias, percentage of scatter points within error envelop (EE%), and root-mean-squared-error (rmse).

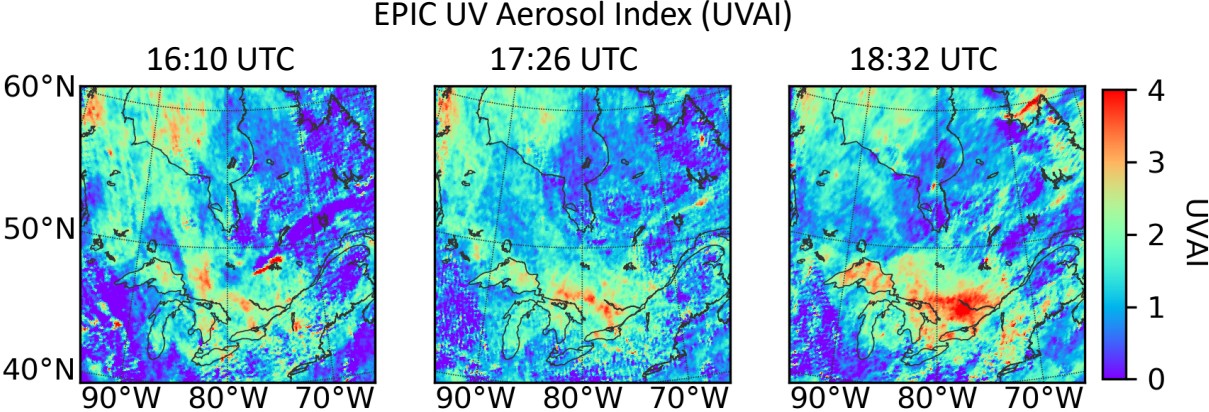

**Figure 11.** EPIC UV aerosol index (UVAI) for the three scenes on 25 August 2017 shown in Figure 7. Here the UVAI was
5   obtained from EPIC Level-2 UV aerosol products.

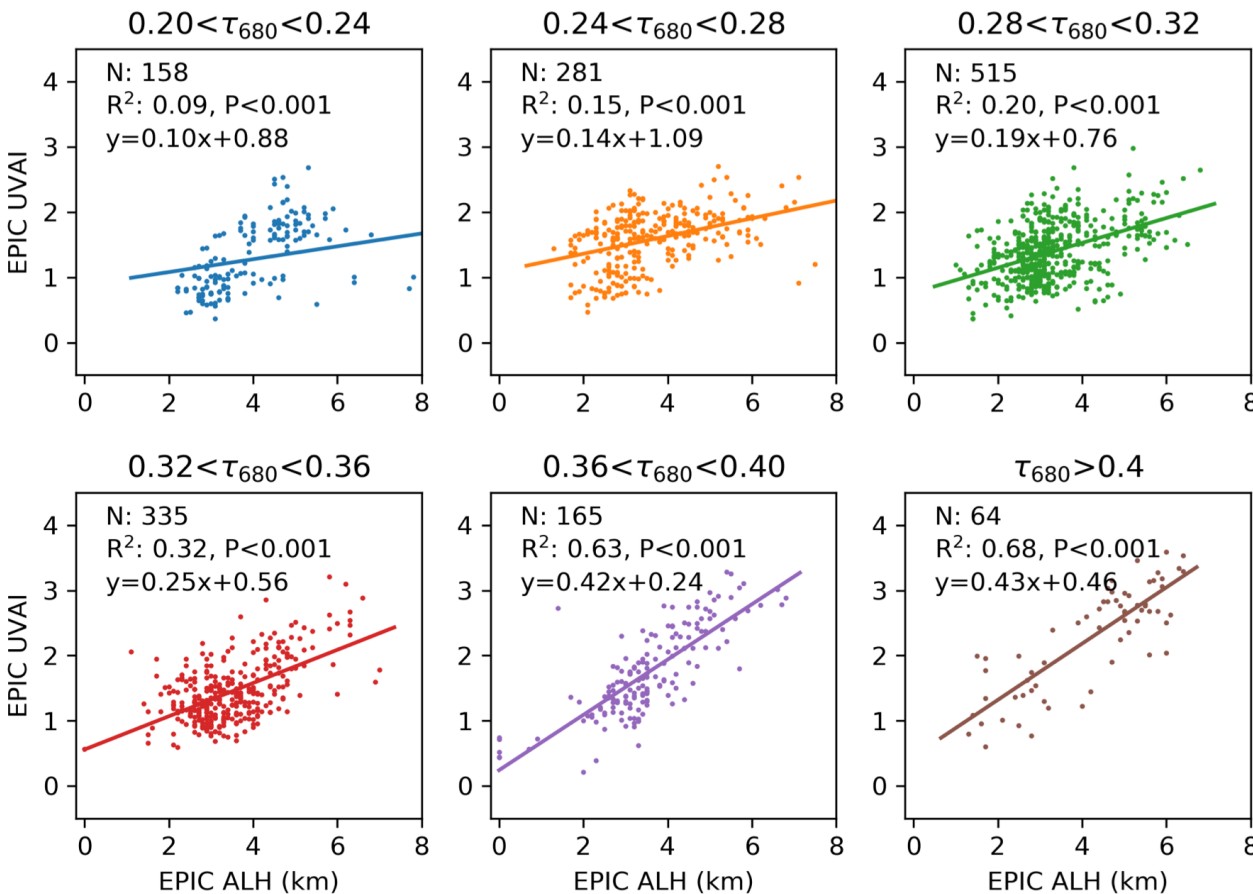

**Figure 12.** Scatterplots of EPIC UVAI (Figure 11) versus current retrievals of ALH (Figure 7) on 25 August 2017 (17:26 UTC) under different AOD values as indicated in the title of each panel. Also annotated are the linear regression fit, number of scatter points (N), coefficient of determination ($R^2$), and significance level (P).

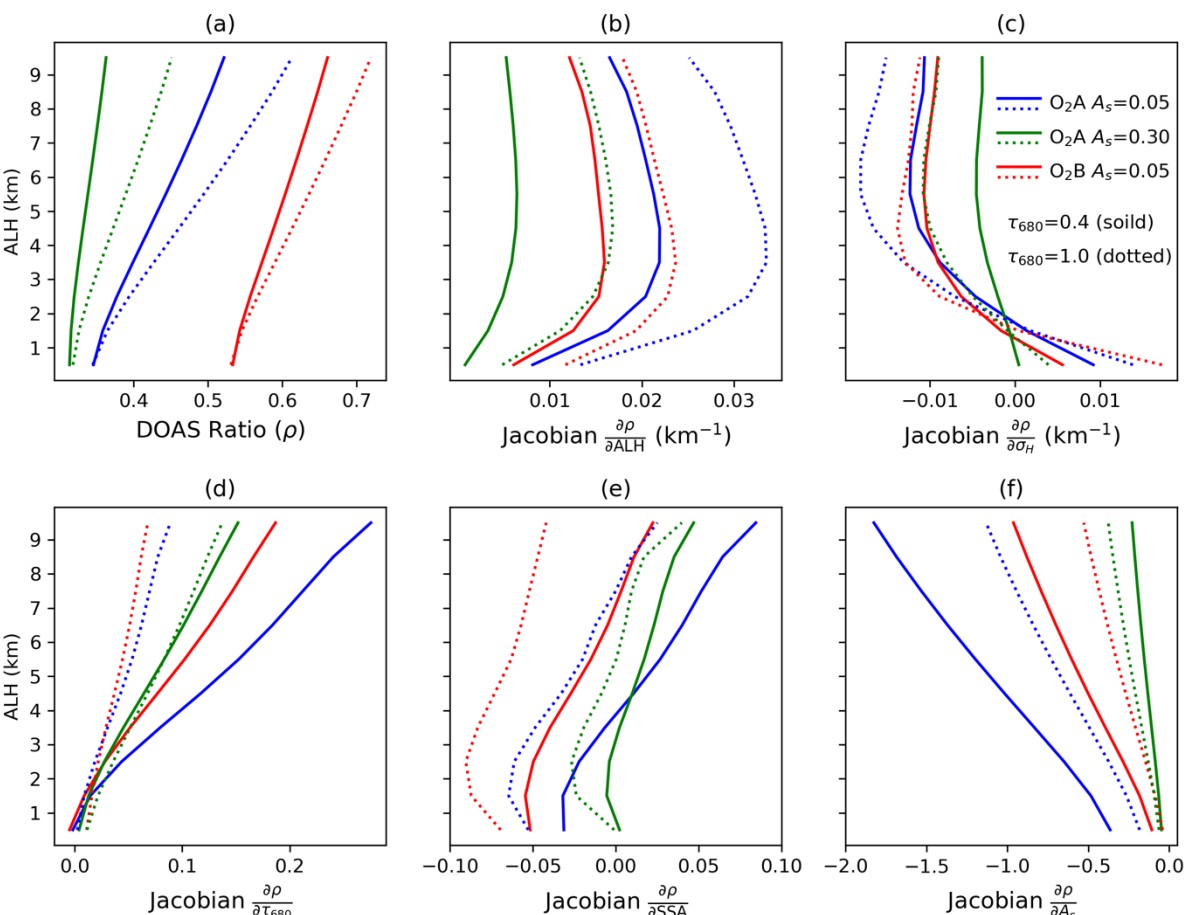

**Figure A1.** UNL-VRTM simulated DOAS ratios $\rho$ in EPIC $O_2$ A and B band (a), and their Jacobian gradients with respect to ALH (b), half-width parameter $\sigma_H$ for the quasi-Gaussian aerosol vertical profile (c), 680-nm AOD $\tau_{680}$ (d), SSA (e), and surface reflectance $A_s$ (f). The y-axis represents aerosols being present at various ALH values. Three colors indicate $\rho$ and its Jacobians for the $O_2$ A band with $A_s = 0.05$ (blue), $O_2$ A band with $A_s = 0.30$ (green), and $O_2$ B band with $A_s = 0.05$ (red). Two AOD loadings are indicated by the solid ($\tau_{680} = 0.4$) and dotted ($\tau_{680} = 1.0$) lines, respectively. Simulations are performed with the same aerosol model and observation geometry in Figure 3.

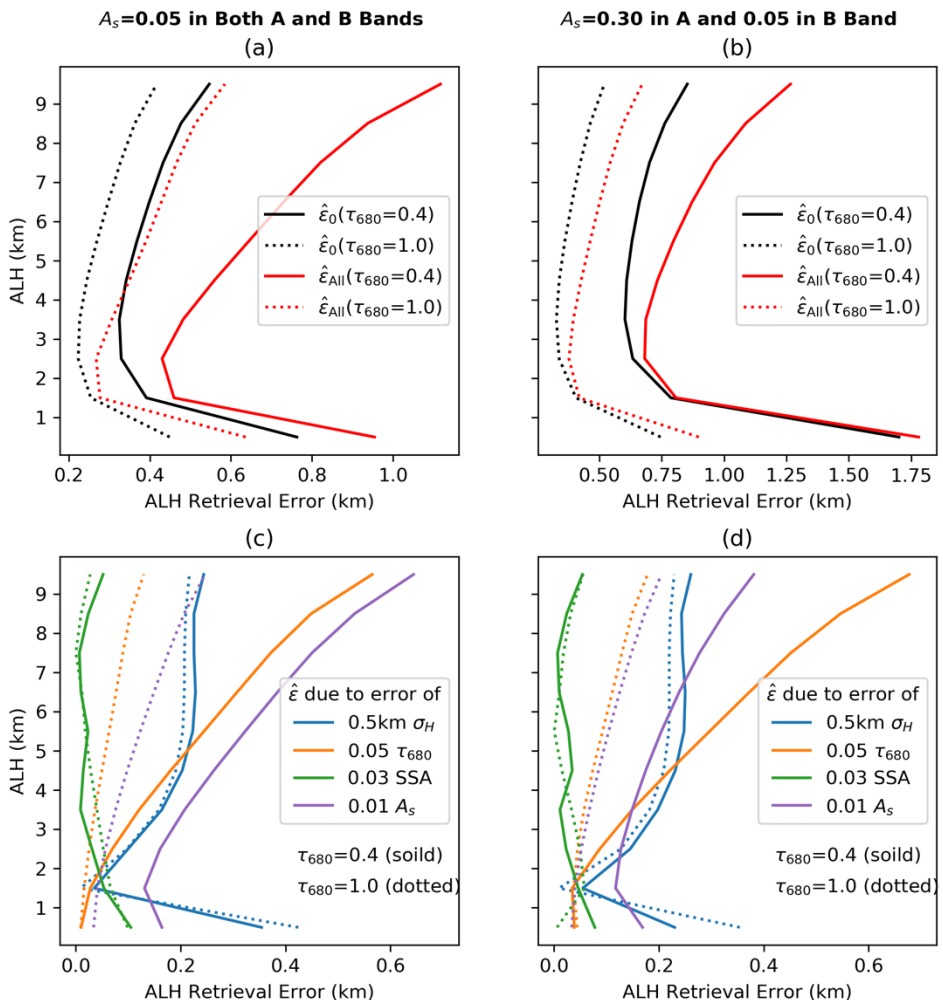

**Figure A2.** Estimated ALH retrieval errors owing to EPIC measurement errors and uncertainties in aerosol and surface assumptions. Left column is for a surface with $A_s = 0.05$ reflectance in both the $O_2$ A and B bands (close to a water surface), and right column is for $A_s = 0.30$ in the A band and $A_s = 0.05$ in the B band (close to a vegetation surface). Top panels show ALH retrieval errors due to 2% EPIC measurement error alone (black curves) and with model uncertainty from all four parameters added (red curves). Bottom panels show the error in retrieved ALH due to uncertainty from each of the parameters: 0.5 km in $\sigma_H$ (blue), 0.05 in $\tau_{680}$ (orange), 0.03 in SSA (green), and 0.01 in $A_s$ (purple). In all panels, the solid and dotted curves represent 0.4 and 1.0 $\tau_{680}$, respectively.