# Peer review of "Detecting layer height of smoke aerosols over vegetated land and water surfaces via oxygen absorption bands: Hourly results from the EPIC/DSCOVR in deep space"

_Atmospheric Measurement Techniques, 2018_

## Referee Comment (RC1) · Anonymous Referee #1 · 15 Jan 2019

Review of the paper "Detecting layer height of smoke aerosols over vegetated land and water surfaces via oxygen absorption bands: Hourly results from EPIC/DSCOVR satellite in deep space" (amt\_2018\_414) by Xu et al.

**General comments**

This paper demonstrates nicely that aerosol layer height information can be retrieved from EPIC/DSCOVR data. This is especially of interest since hourly information can be retrieved – a unique contribution indeed.

The paper should be published following minor revisions which are mostly due to suggested minor phrasing corrections.

**Specific Comments**

Page 7, lines 20-23. These two sentences are awkward. The sentence "Besides, cloud mask thresholds" leaves the reader in a state of uncertainty. The phrase "might need.." is inconclusive. It is suggested to delete the sentence "Besides, cloud mask thresholds". Perhaps one can replace this sentence with "This is a topic of further investigation."

Page 10, lines 10-15. The sentence "To compensate for this bias,.." is not clear. I am having difficulty in accepting the methodology used to account for undetected aerosol. How can one impose an exponentially-decaying background aerosol amount to an undetected aerosol layer if you don't know if the undetected aerosol is there or not? To assume that undetected aerosol is everywhere is problematic. The amount of undetected aerosol likely varies from place to place. Furthermore, the summertime Arctic aerosols do not correspond to conditions elsewhere. I think it would be best to estimate the bias in ALHCALIOP due to the undetected aerosol for a number of observations, state the uncertainty in the paper, and then calculate ALHCALIOP without adding undetected AOD amounts anywhere.

Page 11, lines 28-29. What are typical ALH uncertainties due to MODIS surface products uncertainties and GOME-2 LER uncertainties?

**Technical corrections**

Page 1, line 3: change to "from the EPIC/DSCOVR"

Page 1, line 38: change to "temperature, influence the measured aerosol extinction profiles"

Page 3, line 4: change to "aerosol extinction profiles measured"

Page 3, line 20: change to "ALH utilizing the O2"

Page 6, line 2: change to "where C(ïĄň) is the EPIC"

Page 6, line 29: change to "from analyzing USGS (United States Geological Survey)"

Page 7, line 30: change to "constructed with the UNL-VRTM model."

Page 8, line 1: change to "It also incorporates HITRAN spectroscopic gaseous absorption with up to 22 trace gases"

Page 8, line 27: change to "satellite instrument, separate over-land"

Page 10, line 6: change to "(2013), the CALIOP day tine aerosol extinction threshold is 0.01 - 0.03 km-1 for 80-km horizontal resolution and up to 0.07 km-1 for 5-km horizontal resolution."

Page 10, line 19: change to "65% of the ALH retrievals are within an uncertainty envelope of"

Page 10, line 22: change to "The collocation method follows Ichoku et al. (2002), but was"

Page 10, line 33: change to "smoke by using the UV aerosol"

Page 11, line 9: change to "satellite, since both perform hyperspectral measurements from the UV to the NIR and both cover the O2 A and B bands"

Page 11, line 21: change to "dust ALH from the EPIC experiment (Xu"

Page 11, line 28: change to "information. Surface reflectance values are specified using MODIS"

Page 12, line 5: change to "The three years of data recorded"

Page 12, line 18: change to "group at the University of Iowa"

Page 12, line 20: change to "acknowledge the AERONET program"

Page 20, line 6: change to "includes all ten EPIC bands,"

Page 21, line 8: change to "resulting in less absorption by O2 and "

Page 22, line 4: change to "and surface reflectance (As) values".

Please also note the supplement to this comment: https://www.atmos-meas-tech-discuss.net/amt-2018-414/amt-2018-414-RC1supplement.pdf

---

## Referee Comment (RC2) · Xu (Referee) · 23 Jan 2019

In this paper, Xu et al. extends their earlier work on aerosol layer height (ALH) and aerosol optical depth (AOD) retrieval from ocean to land - using EPIC observations. Comparison to CALIOP retrieval of ALH and AERONET product of AOD show an error of 0.58 km in retrieved ALH and an error of 0.05 in retrieved AOD, respectively. This work was very well organized. I don't have questions on the general technical routines presented in this work as the previous publications by the authors along this line of research have laid down a solid basis for proceeding with this work. My comments

below are for the authors to somehow clarify their approach that readers can better digest the ideas behind it: 1. What is source of oxygen profile adopted in EPIC ALH and AOD retrievals ? Is there any impact from temperature on oxygen profile and then on ALH retrieval ? 2. The sensitivity of O2-A and B bands to the profiles of smoke aerosols are clear from Figure 3. I'm curious how much the sensitivity will change if aerosol absorption changes gradually from being strongly absorbing to being weakly absorbing ? 3. I believe the authors have published it elsewhere, but it would be helpful to some readers if some comments from authors' side can be made on the sensitivity of O2-A and B bands to the width of aerosol layer. 4. To give people a better idea about the "real" sensitivity that measurements have to AOD and ALH, it will be nicer if the authors can describe in the caption of Figure 3 EPIC's measurement errors in the two O2 bands. Probably it would be clear if Z-score (ratio of difference of signals in A/B band signals as normalized by measurement errors) is plotted as its axis. 5. Page 5, though there is a reason (geolocation) on the aggregation of pixels into a box of 3 x 3 pixels, do the authors think the price to pay (reduction of spatial resolution from 24 km to 8 km) too high. What if retrieval is directly implemented on 1 by 1 grid to retain EPIC's original 8 km resolution? 6. Page 8, the smoke particle properties are described for retrieving AOD and ALH. It might helpful if some comments can be given regarding if the pre-determined aerosol model have certain errors and its potential impact on ALH and AOD retrieval accuracy.

Some suggested editorial changes: 1. The reference for Lewis [1994] is not complete. Second author's last name is missing. Correct the citation of this reference in line 24 of Page 6 as well. 2. Delete the redundant set of words "over land" in line 26 of Page 8. 3. Line 27 of Page 8: should "separated" be "separate"?

---

## Referee Comment (RC3) · Anonymous Referee #3 · 24 Jan 2019

General Comments:

This work builds upon the authors' prior work on retrieving aerosol layer height (ALH) over ocean by extending the algorithm to work over land surfaces. One of the most important differences between the ocean and land studies is the characterization of surface reflectance, which is much more variable and complex than that over ocean. Further, their algorithm has a new smoke aerosol model for retrieving biomass-burning smoke ALH. The ALH retrievals are critical for obtaining accurate aerosol products in the UV. They also employ new data aggregation and spectral fitting strategies. Overall,

the work is sound and relevant, and should be published.

Specific Comments:

Since the work focuses on land surfaces, what are the effects of non-Lambertian surface reflection (BRDF) on the retrievals?

Is there a difference in ALH retrieval sensitivity as a function of single scattering albedo?

What are the uncertainties in ALH retrievals due to uncertainties in the surface albedo and pressure climatologies used in the algorithm?

There are a lot of grammatical and typographical errors (way more than acceptable for a manuscript published in this journal) in the manuscript, which I have tried to capture in the technical comments section. These must be corrected before the manuscript can be published.

Technical Comments:

Line 18, page 1: in visible -> in the visible

Line 19, page 1: flexible spectral fittings that account for -> flexible spectral fitting that accounts for

Line 21, page 1: to derive the ALH that represents an optical centroid altitude -> to derive ALH, which represents an optical centroid altitude

Line 22, page 1: the measurements -> measurements

Line 23, page 1: United State -> the United States

Lines 23-24, page 1: the algorithm can well capture -> that the algorithm can be used to obtain

Lines 24-25, page 1: Validations are performed against aerosol extinction profile -> Validation is performed against aerosol extinction profiles

Line 26, page 1: and AOD -> , and against AOD

Line 26, page 1: in average -> on average

Line 29, page 1: the EPIC's -> EPIC's

Line 32, page 1: earth's -> Earth's

Line 34, page 1: arrange the authors alphabetically

Lines 37-38, page 1: The thermal signature of dust in particular can likewise influence the earth longwave budget and through the interference of retrievals of water vapor and temperature, thus influencing measure atmospheric state -> The thermal signature of dust, in particular, can likewise influence the Earth's longwave budget, and through interference with retrievals of water vapor and temperature, influence measurement of the atmospheric state

Line 39, page 1-line 3, page 2: Additionally, the knowledge of ALH is essential in retrieving aerosol absorption properties . . ., in retrieving aerosol microphysical properties . . ., and in the atmospheric correction for ocean color remote sensing -> Additionally, knowledge of ALH is essential for retrieving aerosol absorption properties . . ., aerosol microphysical properties . . ., and for atmospheric correction for ocean color remote sensing

Line 5, page 2: arrange the authors alphabetically; leave a space between the two citations

Line 18, page 2: from UV to near-infrared-> from the UV to the near-infrared

Line 19, page 2: Figure 1b-c -> Figures 1b-c

Line 20, page 2: the spectral -> spectral

Line 22, page 2: present -> presented

Line 23, page 2: the EPIC -> EPIC

Line 24, page 2: demonstrate -> demonstrated

Line 26, page 2: in determining -> for determining

Line 28, page 2: robust strategies in the -> rob ust strategies for the

Line 33, page 2: to retrieving -> for retrieving

Line 33, page 2-line 1, page 3: implicating our algorithm development limited to water and vegetated land surface -> limiting our algorithm development to water and vegetated land surfaces

Line 1, page 3: the assumptions -> assumptions

Lines 2-3, page 3: The ALH retrievals are demonstrated in Section 4, that were applied to smoke events over Canada and the United States in August 2017. -> ALH retrievals of smoke events over Canada and the United States in August 2017 are demonstrated in Section 4.

Line 3, page 3: ALH and AOD from EPIC with -> ALH and AOD from EPIC against

Line 5, page 3: remove "In conclusion,"

Line 11, page 3: Figure 1b-c -> Figures 1b-c

Line 13, page 3: the scattering of presented aerosol particles interact with -> scattered light from aerosol particles interacts with

Line 16, page 3: estimate -> estimated

Line 18, page 3: ' among those are -> ; among those are

Line 20, page 3: with O2 absorption -> using the O2

Line 22, page 3: thus reduce the chance of a photon -> thus reducing the chance of that photon

Line 23, page 3: TOA -> Top Of the Atmosphere (TOA)

Line 28, page 3: leave a space between citations

Lines 29-30, page 3: a state-of-the-art -> the state-of-the-art

Line 31, page 3: formulated by -> for

Line 33, page 3: at the geometry of -> for the geometry

Line 5, page 4: contribution from surface -> contribution from the surface

Line 7, page 4: findings -> the findings; the O2 A and B band -> O2 A and B band

Line 8, page 4: leave a space between citations

Line 11, page 4: the retrieval accuracy -> a retrieval accuracy

Line 13, page 4: earth -> Earth

Line 18, page 4: earth -> Earth

Line 20, page 4: remove the comma

Line 31, page 4: at six EPIC -> in six EPIC

Line 4, page 5: at EPIC bands -> in EPIC bands

Line 5, page 5: EPIC original pixels -> original EPIC pixels

Line 9, page 5: of available pixels -> for the available pixels

Line 11, page 5: specific surface type -> the specific surface type

Lines 12-13, page 5: While the retrieval procedure is based up on our algorithm . . . from the EPIC (Xu et al., 2017), it was upgraded in a few aspects -> While the retrieval procedure is based on our algorithm . . . from EPIC measurements (Xu et al., 2017), it was upgraded in several ways.

Line 13, page 5: algorithm extends -> algorithm is extended

Line 14, page 5: O2 -> O2

Line 24, page 5: Obtain -> Obtaining

Lines 29-30, page 5: The two O2 absorption channels (688 nm and 764 nm) were calibrated by lunar surface reflectivity with EPIC lunar observations at the time of full moon as seen from the earth -> The two O2 absorption channels (688 nm and 764 nm) were calibrated using lunar surface reflectivity from EPIC lunar observations at the time of full moon as seen from Earth

Line 32, page 5: by calibration factors derived by above studies -> using calibration factors from previous studies

Line 34, page 5: top-of-the-atmosphere (TOA) -> TOA

Line 2, page 6: where ðİŘű(ðİlJE) is EPIC measured signal in the units of -> where ðİŘű(ðİlJE) is the EPIC measured signal in units of

Line 6, page 6: Determine -> Determining

Line 8, page 6: leave a space between the citations

Line 12, page 6: leave a space between the citations

Line 16, page 6: in MODIS's first seven channels -> in the first seven MODIS channels; leave a space between the citations

Line 24, page 6: leave a space between the citations

Lines 25-26, page 6: Lambertian surface albedo at MODIS bands of 469, 555, 645, and 858 nm -> Lambertian surface albedo in the 469, 555, 645, and 858 nm MODIS bands

Line 28, page 6: EPIC-bands -> EPIC bands; in the forms -> in the form

Line 31, page 6: spectral locations -> the spectral locations

Line 32, page 6: at each EPIC band -> in each EPIC band; Figure 6c-h -> Figures 6c-h

Line 12, page 7: Mask -> Masking

Line 15, page 7: the land and water -> land and water

Lines 21-22, page 7: higher-resolution geostationary sensors' cloud mask information -> higher resolution cloud mask information from geostationary sensors

Lines 22-23, page 7: if applied operationally -> for operational applications

Line 26, page 7: with MODIS land surface -> using MODIS land surface

Line 30, page 7: constructed with -> constructed using the

Line 3, page 8: of the current retrieval -> for the current retrieval

Line 4, page 8: circumstances -> scenarios

Line 5, page 8: simulated by -> simulated using

Line 6, page 8: at the selected 6 bands -> for the selected six bands

Line 9, page 8: leave a space between the citations

Line 10, page 8: by following -> following

Line 15, page 8: leave a space between the citations

Lines 17-18, page 8: total AOD at the wavelength of 680 nm -> the total AOD at 680 nm

Line 19, page 8: fittings -> fitting

Line 20, page 8: both the water -> both water

Line 21, page 8: fittings -> fitting; account for specifics of surface reflectivity -> accounts for the specifics of surface reflectivity

Lines 21-22, page 8: First, TOA reflectance in EPIC's "atmospheric window" channels are matched with LUTs to determine AOD, because at these channels the TOA reflectance is independent of ALH. -> First, the TOA reflectance in the EPIC "atmospheric window" channels are matched with LUTs to determine AOD, since the TOA reflectance does not depend on ALH in these channels.

Line 26, page 8: because over land the satellite signal tends to be dominated by surface contributions over land -> since the satellite signal tends to be dominated by surface contributions over land

Line 27, page 8: separated -> separate

Line 28, page 8: in characterizing -> for characterizing

Line 30, page 8: the surface type -> surface type

Line 33, page 8: In contrast, the band of 780 nm is excluded for the spectral fitting -> In contrast, the 780 nm band is excluded for spectral fitting

Lines 1-2, page 9: weights to ratios in the O2 A and B bands are given differently for different surfaces -> different weights are given for the ratios in the O2 A and B bands for different surfaces

Line 5, page 9: Demonstration -> demonstration

Line 9, page 9: shown in EPIC RGB images -> shown in the EPIC RGB images

Line 10, page 9: plumes emitted from wildfires in western Canada and, crossing -> plumes emitted from wildfires in western Canada and crossing

Lines 11-12, page 9: The retrieved smoke ALH are shown in Figure 7b and 8b; and retrieved 680-nm AOD in Figure 7c and 8c. -> The retrieved smoke ALH is shown in Figure 7b and 8b, and retrieved 680-nm AOD in Figure 7c and 8c.

Line 13, page 9: and ALH retrievals -> ALH retrievals

Line 18, page 9: towards southeast -> southeast

Line 23, page 9: validations -> validation

Line 25, page 9: observation -> observations

Line 29, page 9: in 532 nm -> at 532 nm

Line 31, page 9: defined in our EPIC algorithm -> as defined in our EPIC algorithm

Line 1, page 10: with the layers where aerosols are detected -> for the layers where aerosols are detected

Line 4, page 10: backscattering ratio that depends -> backscattering ratio, which depends

Line 6, page 10: daytime CALIOP scan -> a daytime CALIOP scan

Line 7, page 10: reaches up to -> increases to

Line 9, page 10: predominately -> predominantly

Lines 10-11, page 10: To compensate for this bias, we use a exponentially-decayed background aerosol extinction profile for substitute of aerosol extinction coefficients of these undetected aerosol layers within troposphere. -> To compensate for this bias, we use an exponentially-decaying background aerosol extinction profile to provide a proxy for aerosol extinction coefficients of these undetected aerosol layers within the troposphere.

Line 13, page 10: summertime atmosphere of the Arctic -> summertime Arctic atmosphere

Line 15, page 10: bias of ALHCALIOP -> bias in ALHCALIOP

Lines 16-17, page 10: Quantitatively, 67

Lines 18-20, page 10: Considering all EPIC- CALIOP ALH pairs, 65

Line 21, page 10: observations of 675 nm AOD -> 675 nm AOD observations

Line 22, page 10: (Ichoku et al., 2002) -> Ichoku et al. (2002)

Lines 22-24, page 10: "but was modified to associate a subset of satellite retrievals within a 3 X 3 AOD subset centered at each site to a subset of 1-hour AERONET observations around EPIC scan time." It is not clear what the authors mean by 3 X 3 AOD subset. The sentence needs to be revised for clarity.

Line 24, page 10: EPIC scan time -> the EPIC scan time

Lines 24-25, page 10: Comparison of EPIC AOD and AERONET are shown in Figure 10b. -> A comparison of EPIC and AERONET AODs is shown in Figure 10b.

Lines 25-26, page 10: The collocated AOD pairs, though with limited data samplings, have over 77

Line 26, page 10: EPIC AOD -> The EPIC AOD

Line 34, page 10: UV aerosol index -> the UV aerosol index

Line 1, page 11: because -> since

Line 9, page 11: both perform -> both of which obtain

Line 10, page 11: leave a space between the citations; Is it Omar et al or Torres et al?

Line 14, page 11: which are in contrast to clouds which-> which are in contrast to clouds that

Line 15, page 11: Because -> Since

Line 16, page 11: correlation -> the correlation

Lines 17-18, page 11: may results in a value UVAI from less than 1 to about 4 -> may result in UVAI values ranging from less than 1 to about 4

Line 19, page 11: EPIC's O2 bands -> the EPIC O2 bands

Lines 21-22, page 11: Based on our previous efforts in retrieving over-water dust ALH from the EPIC (Xu et al., 2017), we extend the retrieval algorithm to biomass burning smoke aerosols over both the water and vegetated land surfaces. -> We extend our retrieval algorithm for retrieving over-water dust ALH from EPIC (Xu et al., 2017) to biomass burning smoke aerosols over both water and vegetated land surfaces.

Line 23, page 11: flexible spectral fittings that account for specifics of-> flexible spectral fitting that accounts for the specifics of

Line 25, page 11: then uses -> and then uses

Line 28, page 11: And, surface reflectance -> Surface reflectance

Lines 31-32, page 11: We found the algorithm captures AOD and ALH multiple times daily over both the water and vegetated land surfaces. -> The algorithm is able to retrieve AOD and ALH multiple times daily over both water and vegetated land surfaces.

Lines 33-34, page 11: , showing EPIC retrieved ALH has a rmse of 0.58 km and captures 52

Line 1, page 12: mrse -> rmse

Lines 1-2, page 12: and over 77

Line 2, page 12: What does an error envelope of +/- (0.05 + 10

Line 4, page 12: the EPIC's UV bands -> the EPIC UV bands

Line 9, page 12: dust or smoke -> (dust or smoke)

Line 15, page 12: NASA the DSCOVR Earth Science Algorithms Program -> the NASA DSCOVR Earth Science Algorithms Program

Line 16, page 12: Office of Naval Research (ONR's) -> the Office of Naval Research (ONR)

Line 17, page 12: under the award -> under award

Line 20, page 12: NASA's -> the NASA; AERONET program -> the AERONET program

Line 21: the AOD data -> AOD data

Table 2 caption: in constructing the LUTs -> for constructing the LUTs

Figure 1 caption: Change to: EPIC instrument filter response function (blue) and atmospheric spectral transmission 5 (orange). Panel (a) includes all ten EPIC bands, whereas panels (b) and (c) show zoom-ins for the 688-nm channel in the O2 B-band and the 764-nm channel in the O2 A-band, respectively. Here, the atmospheric transmission is simulated by the UNL-VRTM model, with a spectral step size and a spectral full width at half maximum of 0.02 nm.

Line 5, page 21: physical principal for -> physical principle of

Line 6, page 21: scattering of aerosol -> scattering by aerosol

Line 7, page 21: path way -> pathlength

Line 8, page 21: than in the lower-altitude aerosol -> than those scattered by the lower-altitude aerosol; less chance -> lower chance

Line 5, page 22: at the geometry of -> for the geometry

Figure 4 legend: Change "Green vegetations" to "Green vegetation surfaces"

Figure 5: What does 3X3 aggregation mean? Do you aggregate 9 pixels at a time? Why? Some explanation is needed in the text and better wording in the Figure.

Line 5, page 25: the statistics -> statistics

Line 6, page 25: red dot line -> red dotted lines; their respective -> the respective

Lines 7-8, page 25: reflectance at each EPIC band versus reflectance at corresponding MODIS bands -> reflectance in each EPIC band versus reflectance in the corresponding MODIS bands

Line 4, page 26: UTC time -> UTC times

Line 6, page 26: CALIOP sub-orbital track with an overpass time 19:05 UTC -> the CALIOP sub-orbital track with an overpass time of 19:05 UTC

Line 8, page 26: EPIC scan time -> the EPIC scan time

Line 5, page 27: UTC time -> UTC times

Line 6, page 27: CALIOP overpass time was at 18:15. -> The CALIOP overpass was at 18:15 UTC.

Line 4, page 28: Comparison of ALH retrieved from EPIC and the ALH derived from CALIOP level-2 aerosol extinction profile -> Comparison of ALH retrieved from EPIC and CALIOP level-2 aerosol extinction profile

Line 5-6, page 28: CALIOP orbital tracks are marked on EPIC RGB images in Figure 7–8. -> The CALIOP orbital tracks are marked on EPIC RGB images in Figures 7–8.

Line 7-8, page 28: Error bar of EPIC ALH represents standard deviation for an array of 3x3 24-km retrieval pixels, while the error bar of CALIOP ALH represents standard deviation of over 5 adjacent CALIOP 5-km footprints. -> The error bar for EPIC ALH represents the standard deviation for an array of 3x3 24-km retrieval pixels, while that for CALIOP ALH represents the standard deviation of over 5 adjacent CALIOP 5-km footprints.

Line 4, page 29: counterparts from -> corresponding

Line 5, page 29: Color of -> The color of

Line 6, page 29: scatter -> scatter point; EPIC 680-nm AOD value of -> the EPIC 680-nm value for

Line 7, page 29: Dotted lines -> The dotted lines

Line 8, page 29: one-by-one -> the one-to-one

Line 9, page 29: regression fitting -> regression fit; scatters -> scatter points; the linear -> linear

Line 10, page 29: scatters -> scatter points

Line 4, page 30: UVAI were -> UVAI was

Line 4, page 31: linear regression fitting -> the linear regression fit

Line 5, page 31: scatters -> scatter points

---

## Author Comment (AC1) · 7 May 2019

We thank the reviewer for his/her careful reading of the manuscript and offering many constructive feedbacks and helpful suggestions. We have incorporated most of the suggestions and believe the revised paper is substantially improved. In order for the reviewers and the editor to more readily identify our changes, we've submitted two versions of the revised paper, one with "track changes" and the other with same changes incorporated.

Because we made substantial revisions, we include below a list of main changes, after which we provide a detailed response to the reviewer's comments. The original comments by the reviewer are in black font, our replies in blue.

**Major changes in the revised manuscript**:

- We added an appendix to investigate the potential retrieval error in ALH due to several assumptions made in the retrieval algorithm, including the surface reflectance, smoke single scattering albedo, aerosol optical depth, and half width of the assumed quasi-Gaussian aerosol vertical profile.
- In the validation for EPIC-retrieved ALH, we now use two sets of CALIOP-based ALH (updated Figures 9 and 10): one with a background aerosol amount added for undetected CALIOP aerosol layers, the other one without. While there is a mean difference of 0.36 km between these two sets of CALIOP ALHs, we found EPIC ALH retrievals are in general consistent with the both sets with a RMSE of 0.57-0.58 km.
- The language of the manuscript has been substantially improved by incorporating reviewers' comments and authors' further proof reading.

**Anonymous Referee #1**

This paper demonstrates nicely that aerosol layer height information can be retrieved from EPIC/DSCOVR data. This is especially of interest since hourly information can be

retrieved – a unique contribution indeed. The paper should be published following minor revisions which are mostly due to suggested minor phrasing corrections.

We thank the reviewer's positive comments to the significance of this article.

Page 7, lines 20-23. These two sentences are awkward. The sentence "Besides, cloud mask thresholds" leaves the reader in a state of uncertainty. The phrase "might need.." is inconclusive. It is suggested to delete the sentence "Besides, cloud mask thresholds". Perhaps one can replace this sentence with "This is a topic of further investigation."

Following the reviewer's suggestion, we removed the sentence "Besides, cloud mask thresholds … operational applications."

Page 10, lines 10-15. The sentence "To compensate for this bias,.." is not clear. I am having difficulty in accepting the methodology used to account for undetected aerosol. How can one impose an exponentially-decaying background aerosol amount to an un-detected aerosol layer if you don't know if the undetected aerosol is there or not? To assume that undetected aerosol is everywhere is problematic. The amount of undetected aerosol likely varies from place to place. Furthermore, the summertime Arctic aerosols do not correspond to conditions elsewhere. I think it would be best to estimate the bias in ALHCALIOP due to the undetected aerosol for a number of observations, state the uncertainty in the paper, and then calculate ALHCALIOP without adding undetected AOD amounts anywhere.

Thank you for this suggestion. Through a close inspection, we found that the background aerosol profile was not added in the CALIOP ALH shown in Figures 9 and 10a. So, in the revised manuscript, we keep those figures based on your suggestion. At the same time, in the validation analysis we also included the CALIOP ALH by adding undetected background aerosol. We found the mean difference between these two sets of CALIOP ALH was 0.36 km, and our EPIC ALH retrievals are consistent with both sets

of CALIOP ALH with a RMSE of 0.57-0.58 km (see the revised Figures 9 and 10 in the revised manuscript).

Page 11, lines 28-29. What are typical ALH uncertainties due to MODIS surface products uncertainties and GOME-2 LER uncertainties?

To answer this question, we added an error analysis and discussion (in the article Appendix) for ALH retrievals due to uncertainties in surface reflectance and other parameters. We found a surface reflectance error of 0.01 may lead to ALH retrieval errors from 0.1 km to 0.6 km for both water and vegetation surface types, depending on the aerosol loading and vertical allocation.

Technical corrections:

Page 1, line 3: change to "from the EPIC/DSCOVR"

Corrected.

Page 1, line 38: change to "temperature, influence the measured aerosol extinction profiles"

Corrected.

Page 3, line 4: change to "aerosol extinction profiles measured"

Corrected.

Page 3, line 20: change to "ALH utilizing the O2"

We changed to "ALH using the O2".

Page 6, line 2: change to "where C($\lambda$) is the EPIC"

Corrected.

Page 6, line 29: change to "from analyzing USGS (United States Geological Survey)"

Corrected.

Page 7, line 30: change to "constructed with the UNL-VRTM model."

We changed to "constructed using the UNL-VRTM model."

Page 8, line 1: change to "It also incorporates HITRAN spectroscopic gaseous absorption with up to 22 trace gases"

Corrected.

Page 8, line 27: change to "satellite instrument, separate over-land"

Corrected.

Page 10, line 6: change to "(2013), the CALIOP day tine aerosol extinction threshold is 0.01 – 0.03 km-1 for 80-km horizontal resolution and up to 0.07 km-1 for 5-km horizontal resolution."

We changed to "(2013), aerosol extinction threshold in a daytime CALIOP scan is 0.01 – 0.03 km–1 for 80-km horizontal averaging resolution and increases to 0.07 km–1 for 5-km horizontal averaging resolution."

Page 10, line 19: change to "65% of the ALH retrievals are within an uncertainty envelope of"

Corrected.

Page 10, line 22: change to "The collocation method follows Ichoku et al. (2002), but was"

Corrected.

Page 10, line 33: change to "smoke by using the UV aerosol"

Corrected.

Page 11, line 9: change to "satellite, since both perform hyperspectral measurements from the UV to the NIR and both cover the O2 A and B bands"

We changed to "satellite, both of which obtain hyperspectral measurements from the UV to the NIR covering the O2 A and B bands"

Page 11, line 21: change to "dust ALH from the EPIC experiment (Xu"

We changed to "dust ALH from the EPIC experiments"

Page 11, line 28: change to "information. Surface reflectance values are specified using MODIS"

Corrected.

Page 12, line 5: change to "The three years of data recorded" Page 12, line 18: change to "group at the University of Iowa"

Corrected.

Page 12, line 20: change to "acknowledge the AERONET program"

Corrected.

Page 20, line 6: change to "includes all ten EPIC bands,"

Corrected.

Page 21, line 8: change to "resulting in less absorption by O2 and " Page 22, line 4: change to "and surface reflectance (As) values".

Corrected.

---

## Author Comment (AC2)

We thank Dr. Feng Xu for his careful reading of the manuscript and offering many constructive feedbacks and helpful suggestions. We have incorporated most of the suggestions and believe the revised paper is substantially improved. In order for the reviewers and the editor to more readily identify our changes, we've submitted two versions of the revised paper, one with "track changes" and the other with same changes incorporated.

Because we made substantial revisions, we include below a list of main changes, after which we provide a detailed response to the reviewer's comments. The original comments by the reviewer are in black font, our replies in blue.

**Major changes in the revised manuscript**:

- We added an appendix to investigate the potential retrieval error in ALH due to several assumptions made in the retrieval algorithm, including the surface reflectance, smoke single scattering albedo, aerosol optical depth, and half width of the assumed quasi-Gaussian aerosol vertical profile.
- In the validation for EPIC-retrieved ALH, we now use two sets of CALIOP-based ALH (updated Figures 9 and 10): one with a background aerosol amount added for undetected CALIOP aerosol layers, the other one without. While there is a mean difference of 0.36 km between these two sets of CALIOP ALHs, we found EPIC ALH retrievals are in general consistent with the both sets with a RMSE of 0.57-0.58 km.
- The language of the manuscript has been substantially improved by incorporating reviewers' comments and authors' further proof reading.

**Dr. Feng Xu (Referee #2)**

In this paper, Xu et al. extends their earlier work on aerosol layer height (ALH) and aerosol optical depth (AOD) retrieval from ocean to land - using EPIC observations. Comparison to CALIOP retrieval of ALH and AERONET product of AOD show an error

of 0.58 km in retrieved ALH and an error of 0.05 in retrieved AOD, respectively. This work was very well organized. I don't have questions on the general technical routines presented in this work as the previous publications by the authors along this line of research have laid down a solid basis for proceeding with this work.

We thank the reviewer's positive comments to the significance of this article.

My comments below are for the authors to somehow clarify their approach that readers can better digest the ideas behind it:

1. What is source of oxygen profile adopted in EPIC ALH and AOD retrievals ? Is there any impact from temperature on oxygen profile and then on ALH retrieval ?

The O2 mixing ratio profile is obtained from the mid-latitude standard atmosphere (McClatchey et al., 1972), with surface pressure enlisted from MERRA-2 data. O2 is well mixed in the atmosphere. O2 absorption cross sections are calculated from the HITRAN spectroscopic line parameters in a very high spectral resolution (see Table 2). The high-spectral radiance data then convolved to EPIC bands. Although temperature and pressure have influence to the line position and line width of spectral O2 absorption. Such influence tends to be negligible in the convolved spectral radiance. So, we don't think the there is any impact from temperature on oxygen profile and ALH retrieval.

We have clarified this in section 3.3: "…. Here, we enlisted surface pressure information from the Modern-Era Retrospective analysis for Research and Applications Version 2 (MERRA-2) datasets (Gelaro et al., 2017). MERRA-2's 1-hourly surface pressure at 0.5° by 0.675° grids were interpolated to the location and scan time of each EPIC pixel. In addition, the atmospheric temperature-pressure profile also impacts the width and strength of $O_2$ absorption lines. However, such influence on the radiative transfer is negligible for EPIC's 1-to-2-nm-wide bands. In this study, our algorithm employs a

standard temperature-pressure profile representing the mid-latitude-summer atmosphere (McClatchey et al., 1972)."

2. The sensitivity of O2-A and B bands to the profiles of smoke aerosols are clear from Figure 3. I'm curious how much the sensitivity will change if aerosol absorption changes gradually from being strongly absorbing to being weakly absorbing?

We performed sensitivity and error analysis to answer this question in the Appendix. We found the DAOS ratios used for ALH retrieval are sensitive to SSA, especially for large AOD values (see Figure A1e). However, SSA only has marginal impact to the ALH retrieval error (see Figure A2).

3. I believe the authors have published it elsewhere, but it would be helpful to some readers if some comments from authors' side can be made on the sensitivity of O2-A and B bands to the width of aerosol layer.

In our retrieval algorithm we fixed the profile half-width as I km, because EPIC measurements are not able to resolve both the ALH and half-width at the same time. In the added appendix, we also performed sensitivity and error analysis to answer this question. We found the DAOS ratios used for ALH retrieval are sensitive to the profile half-with, especially for elevated aerosols (see Figure A1c). From the error analysis, we found a change of 0.5 km in the half-width may cause up to 0.3 km error in the retrieved ALH (Figure A2).

4. To give people a better idea about the "real" sensitivity that measurements have to AOD and ALH, it will be nicer if the authors can describe in the caption of Figure 3 EPIC's measurement errors in the two O2 bands. Probably it would be clear if Z-score

(ratio of difference of signals in A/B band signals as normalized by measurement errors) is plotted as its axis.

We thank the reviewer for this suggestion. We added that uncertainty in EPIC DOAS ratios are within 3%. In addition, we added Jacobian of DOAS ratio to ALH, as well as the ALH retrieval error due to the EPIC measurements error, in the appendix. We hope those can give readers a better idea of about the real sensitivity.

5. Page 5, though there is a reason (geolocation) on the aggregation of pixels into a box of 3 x 3 pixels, do the authors think the price to pay (reduction of spatial resolution from 24 km to 8 km) too high. What if retrieval is directly implemented on 1 by 1 grid to retain EPIC's original 8 km resolution?

We indeed have tried the retrieval with native EPIC resolution, which is shown in the figure below for the same smoke case on August 25 in the article. The only reason we do 3x3 pixel aggregation is the EPIC geolocation. I understand EPIC version 3 data with better geolocation and calibration will be released soon. We have a plan to implement the native-resolution retrievals with the new EPIC data.

[Figure]

6. Page 8, the smoke particle properties are described for retrieving AOD and ALH. It might helpful if some comments can be given regarding if the pre-determined aerosol model have certain errors and its potential impact on ALH and AOD retrieval accuracy.

Sure. In the added appendix, we included the analysis and discussion about the sensitivity of our retrieval algorithm and the impacts for ALH retrievals from uncertainties in AOD and in the assumed aerosol model (such as SSA and the Gaussian profile width).

Some suggested editorial changes:

1. The reference for Lewis [1994] is not complete. Second author's last name is missing. Correct the citation of this reference in line 24 of Page 6 as well.

The reference is corrected as:

Lewis, P., and Barnsley, M. J.: Influence of the sky radiance distribution on various formulations of the earth surface albedo, in: Proc. Conf. Phys. Meas. Sign. Remote Sens, Val d'Isere, France, 707-715, 1994.

And the citation for this reference is changed to "Lewis and Barnsley (1994)"

2. Delete the redundant set of words "over land" in line 26 of Page 8.

Corrected.

3. Line 27 of Page 8: should "separated" be "separate"?

Corrected.

---

## Author Comment (AC3)

We thank the reviewer for his/her careful reading of the manuscript and offering many constructive feedbacks and helpful suggestions. We have incorporated most of the suggestions and believe the revised paper is substantially improved. In order for the reviewers and the editor to more readily identify our changes, we've submitted two versions of the revised paper, one with "track changes" and the other with same changes incorporated.

Because we made substantial revisions, we include below a list of main changes, after which we provide a detailed response to the reviewer's comments. The original comments by the reviewer are in black font, our replies in blue.

**Major changes in the revised manuscript**:

- We added an appendix to investigate the potential retrieval error in ALH due to several assumptions made in the retrieval algorithm, including the surface reflectance, smoke single scattering albedo, aerosol optical depth, and half width of the assumed quasi-Gaussian aerosol vertical profile.
- In the validation for EPIC-retrieved ALH, we now use two sets of CALIOP-based ALH (updated Figures 9 and 10): one with a background aerosol amount added for undetected CALIOP aerosol layers, the other one without. While there is a mean difference of 0.36 km between these two sets of CALIOP ALHs, we found EPIC ALH retrievals are in general consistent with the both sets with a RMSE of 0.57-0.58 km.
- The language of the manuscript has been substantially improved by incorporating reviewers' comments and authors' further proof reading.

**Anonymous Referee #3**

General Comments:

This work builds upon the authors' prior work on retrieving aerosol layer height (ALH) over ocean by extending the algorithm to work over land surfaces. One of the most important differences between the ocean and land studies is the characterization of surface reflectance, which is much more variable and complex than that over ocean. Further, their algorithm has a new smoke aerosol model for retrieving biomass-burning smoke ALH. The ALH retrievals are critical for obtaining accurate aerosol products in the UV. They also employ new data aggregation and spectral fitting strategies. Overall, the work is sound and relevant, and should be published.

We thank the reviewer's positive comments to the significance of this article.

Specific Comments:

Since the work focuses on land surfaces, what are the effects of non-Lambertian surface reflection (BRDF) on the retrievals?

By neglecting BRDF effects in the surface reflectance, the ALH retrieval could be biased. Further studies are needed to examine the detailed impacts. On the other hand, since EPIC measurements are mostly in the back-scattering direction, the impacts from BRDF could be limited. This will be one of our future efforts to improve the ALH retrieval accuracy.

So we added following text in section 3.3 to clarify this: "It should be noted that the effects of non-Lambertian surface reflection may bias the ALH retrieval, although this type of impact is limited by EPIC's backscattering direction only observation pattern. Further studies are needed to examine the detailed impacts, which will be one of our future efforts."

Is there a difference in ALH retrieval sensitivity as a function of single scattering albedo?

We performed sensitivity and error analysis to answer this question in the Appendix. We found the DAOS ratios used for ALH retrieval are sensitive to SSA, especially for large

AOD values (see Figure A1e). However, SSA only has marginal impact to the ALH retrieval error (see Figure A2).

What are the uncertainties in ALH retrievals due to uncertainties in the surface albedo and pressure climatologies used in the algorithm?

The uncertainties in ALH retrievals due to uncertainties in surface albedo are analyzed and discussed in the added Appendix. We found a surface reflectance error of 0.01 may lead to ALH retrieval errors from 0.1 km to 0.6 km for both water and vegetation surface types, depending on the aerosol loading and vertical allocations.

We indeed use surface pressure from MERRA-2 reanalysis data, applied to the high-latitude summer standard atmosphere for atmospheric pressure levels. Since, $O_2$ is well mixed the atmosphere, we believe the use of MERRA-2 surface pressure can well capture the amount of $O_2$, and thus has negligible uncertainty due to the atmospheric pressure.

There are a lot of grammatical and typographical errors (way more than acceptable for a manuscript published in this journal) in the manuscript, which I have tried to capture in the technical comments section. These must be corrected before the manuscript can be published.

We really thankful to the reviewer for his/her careful reading of the manuscript and providing very detailed technical comments. We have corrected all of these comments as listed below.

Technical Comments:

Line 18, page 1: in visible -> in the visible

Corrected.

Line 19, page 1: flexible spectral fittings that account for -> flexible spectral fitting that accounts for

Corrected.

Line 21, page 1: to derive the ALH that represents an optical centroid altitude -> to derive ALH, which represents an optical centroid altitude

Corrected.

Line 22, page 1: the measurements -> measurements Line 23, page 1: United State -> the United States

Corrected.

Lines 23-24, page 1: the algorithm can well capture -> that the algorithm can be used to obtain

Corrected.

Lines 24-25, page 1: Validations are performed against aerosol extinction profile -> Validation is performed against aerosol extinction profiles

Corrected.

Line 26, page 1: and AOD -> , and against AOD Line 26, page 1: in average -> on average

Corrected.

Line 29, page 1: the EPIC's -> EPIC's

Corrected.

Line 32, page 1: earth's -> Earth's

Corrected.

Line 34, page 1: arrange the authors alphabetically

Done.

Lines 37-38, page 1: The thermal signature of dust in particular can likewise influence the earth longwave budget and through the interference of retrievals of water vapor and temperature, thus influencing measure atmospheric state -> The thermal signature of dust, in particular, can likewise influence the Earth's longwave budget, and through interference with retrievals of water vapor and temperature, influence measurement of the atmospheric state

Done.

Line 39, page 1-line 3, page 2: Additionally, the knowledge of ALH is essential in re-trieving aerosol absorption properties . . ., in retrieving aerosol microphysical properties . . ., and in the atmospheric correction for ocean color remote sensing -> Additionally, knowledge of ALH is essential for retrieving aerosol absorption properties . . ., aerosol microphysical properties ..., and for atmospheric correction for ocean color remote sensing

Done.

Line 5, page 2: arrange the authors alphabetically; leave a space between the two citations

Done.

Line 18, page 2: from UV to near-infrared-> from the UV to the near-infrared

Corrected.

Line 19, page 2: Figure 1b-c -> Figures 1b-c

Corrected.

Line 20, page 2: the spectral -> spectral

Corrected.

Line 22, page 2: present -> presented

Corrected.

Line 23, page 2: the EPIC -> EPIC

Corrected.

Line 24, page 2: demonstrate -> demonstrated

Corrected.

Line 26, page 2: in determining -> for determining

Corrected.

Line 28, page 2: robust strategies in the -> robust strategies for the

Corrected.

Line 33, page 2: to retrieving -> for retrieving

Corrected.

Line 33, page 2-line 1, page 3: implicating our algorithm development limited to water and vegetated land surface -> limiting our algorithm development to water and vegetated land surfaces

Done.

Line 1, page 3: the assumptions -> assumptions

Corrected.

Lines 2-3, page 3: The ALH retrievals are demonstrated in Section 4, that were applied to smoke events over Canada and the United States in August 2017. -> ALH retrievals of smoke events over Canada and the United States in August 2017 are demonstrated in Section 4.

Corrected.

Line 3, page 3: ALH and AOD from EPIC with -> ALH and AOD from EPIC against

Corrected.

Line 5, page 3: remove "In conclusion,"

Corrected.

Line 11, page 3: Figure 1b-c -> Figures 1b-c

Corrected.

Line 13, page 3: the scattering of presented aerosol particles interact with -> scattered light from aerosol particles interacts with

Corrected.

Line 16, page 3: estimate -> estimated

Corrected.

Line 18, page 3: ' among those are -> ; among those are

Corrected.

Line 20, page 3: with O2 absorption -> using the O2

Corrected.

Line 22, page 3: thus reduce the chance of a photon -> thus reducing the chance of that photon

Corrected.

Line 23, page 3: TOA -> Top Of the Atmosphere (TOA)

Corrected.

Line 28, page 3: leave a space between citations

Done.

Lines 29-30, page 3: a state-of-the-art -> the state-of-the-art

Corrected.

Line 31, page 3: formulated by -> for

Corrected.

Line 33, page 3: at the geometry of -> for the geometry

Corrected.

Line 5, page 4: contribution from surface -> contribution from the surface

Corrected.

Line 7, page 4: findings -> the findings; the O2 A and B band -> O2 A and B band

Corrected.

Line 8, page 4: leave a space between citations

Done.

Line 11, page 4: the retrieval accuracy -> a retrieval accuracy

Corrected.

Line 13, page 4: earth -> Earth

Corrected.

Line 18, page 4: earth -> Earth

Corrected.

Line 20, page 4: remove the comma

Done.

Line 31, page 4: at six EPIC -> in six EPIC

Corrected.

Line 4, page 5: at EPIC bands -> in EPIC bands

Corrected.

Line 5, page 5: EPIC original pixels -> original EPIC pixels

Corrected.

Line 9, page 5: of available pixels -> for the available pixels

Corrected.

Line 11, page 5: specific surface type -> the specific surface type

Corrected.

Lines 12-13, page 5: While the retrieval procedure is based up on our algorithm . . . from the EPIC (Xu et al., 2017), it was upgraded in a few aspects -> While the retrieval procedure is based on our algorithm . . . from EPIC measurements (Xu et al., 2017), it was upgraded in several ways.

Done.

Line 13, page 5: algorithm extends -> algorithm is extended

Corrected.

Line 14, page 5: O2 -> $O_2$

Corrected.

Line 24, page 5: Obtain -> Obtaining

Corrected.

Lines 29-30, page 5: The two O2 absorption channels (688 nm and 764 nm) were calibrated by lunar surface reflectivity with EPIC lunar observations at the time of full moon as seen from the earth -> The two O2 absorption channels (688 nm and 764 nm) were calibrated using lunar surface reflectivity from EPIC lunar observations at the time of full moon as seen from Earth

Corrected.

Line 32, page 5: by calibration factors derived by above studies -> using calibration factors from previous studies

Corrected.

Line 34, page 5: top-of-the-atmosphere (TOA) -> TOA

Corrected.

Line 2, page 6: where $C(\lambda)$ is EPIC measured signal in the units of -> where $C(\lambda)$ is the EPIC measured signal in units of

Corrected.

Line 6, page 6: Determine -> Determining

Corrected.

Line 8, page 6: leave a space between the citations

Done.

Line 12, page 6: leave a space between the citations

Done.

Line 16, page 6: in MODIS's first seven channels -> in the first seven MODIS channels; leave a space between the citations

Corrected.

Line 24, page 6: leave a space between the citations

Done.

Lines 25-26, page 6: Lambertian surface albedo at MODIS bands of 469, 555, 645, and 858 nm -> Lambertian surface albedo in the 469, 555, 645, and 858 nm MODIS bands

Corrected.

Line 28, page 6: EPIC-bands -> EPIC bands; in the forms -> in the form

Corrected.

Line 31, page 6: spectral locations -> the spectral locations

Corrected.

Line 32, page 6: at each EPIC band -> in each EPIC band; Figure 6c-h -> Figures 6c-h

Corrected.

Line 12, page 7: Mask -> Masking

Corrected.

Line 15, page 7: the land and water -> land and water

Corrected.

Lines 21-22, page 7: higher-resolution geostationary sensors' cloud mask information ->
higher resolution cloud mask information from geostationary sensors

Corrected.

Lines 22-23, page 7: if applied operationally -> for operational applications

We removed this sentence "Besides, cloud mask thresholds used in this work might
need to be adjusted if applied operationally."

Line 26, page 7: with MODIS land surface -> using MODIS land surface

Corrected.

Line 30, page 7: constructed with -> constructed using the

Corrected.

Line 3, page 8: of the current retrieval -> for the current retrieval

Corrected.

Line 4, page 8: circumstances -> scenarios

Corrected.

Line 5, page 8: simulated by -> simulated using

Corrected.

Line 6, page 8: at the selected 6 bands -> for the selected six bands

Corrected.

Line 9, page 8: leave a space between the citations

Done.

Line 10, page 8: by following -> following

Corrected.

Line 15, page 8: leave a space between the citations

Done.

Lines 17-18, page 8: total AOD at the wavelength of 680 nm -> the total AOD at 680 nm

Corrected.

Line 19, page 8: fittings -> fitting

Corrected.

Line 20, page 8: both the water -> both water

Corrected.

Line 21, page 8: fittings -> fitting; account for specifics of surface reflectivity -> accounts for the specifics of surface reflectivity

Corrected.

Lines 21-22, page 8: First, TOA reflectance in EPIC's "atmospheric window" channels are matched with LUTs to determine AOD, because at these channels the TOA reflectance is independent of ALH. -> First, the TOA reflectance in the EPIC "atmospheric window" channels are matched with LUTs to determine AOD, since the TOA reflectance does not depend on ALH in these channels.

Corrected.

Line 26, page 8: because over land the satellite signal tends to be dominated by surface contributions over land -> since the satellite signal tends to be dominated by surface contributions over land

Corrected.

Line 27, page 8: separated -> separate

Corrected.

Line 28, page 8: in characterizing -> for characterizing

Corrected.

Line 30, page 8: the surface type -> surface type

Corrected.

Line 33, page 8: In contrast, the band of 780 nm is excluded for the spectral fitting -> In contrast, the 780 nm band is excluded for spectral fitting

Corrected.

Lines 1-2, page 9: weights to ratios in the O2 A and B bands are given differently for different surfaces -> different weights are given for the ratios in the O2 A and B bands for different surfaces

Corrected.

Line 5, page 9: Demonstration -> demonstration

Corrected.

Line 9, page 9: shown in EPIC RGB images -> shown in the EPIC RGB images

Corrected.

Line 10, page 9: plumes emitted from wildfires in western Canada and, crossing -> plumes emitted from wildfires in western Canada and crossing

We changed to "plumes emitted from wildfires in western Canada and transported"

Lines 11-12, page 9: The retrieved smoke ALH are shown in Figure 7b and 8b; and retrieved 680-nm AOD in Figure 7c and 8c. -> The retrieved smoke ALH is shown in Figure 7b and 8b, and retrieved 680-nm AOD in Figure 7c and 8c.

Corrected.

Line 13, page 9: and ALH retrievals -> ALH retrievals

Corrected.

Line 18, page 9: towards southeast -> southeast

Corrected.

Line 23, page 9: validations -> validation

Corrected.

Line 25, page 9: observation -> observations

Corrected.

Line29, page9: in 532 nm -> at 532 nm

Corrected.

Line 31, page 9: defined in our EPIC algorithm -> as defined in our EPIC algorithm

Corrected.

Line 1, page 10: with the layers where aerosols are detected -> for the layers where aerosols are detected

Corrected.

Line 4, page 10: backscattering ratio that depends -> backscattering ratio, which depends

Corrected.

Line 6, page 10: daytime CALIOP scan -> a daytime CALIOP scan

Corrected.

Line 7, page 10: reaches up to -> increases to

Corrected.

Line 9, page 10: predominately -> predominantly

Corrected.

Lines 10-11, page 10: To compensate for this bias, we use a exponentially-decayed background aerosol extinction profile for substitute of aerosol extinction coefficients of these undetected aerosol layers within troposphere. -> To compensate for this bias, we use an exponentially-decaying background aerosol extinction profile to provide a proxy for aerosol extinction coefficients of these undetected aerosol layers within the troposphere.

Corrected.

Line 13, page 10: summertime atmosphere of the Arctic -> summertime Arctic atmosphere

Corrected.

Line 15, page 10: bias of ALHCALIOP -> bias in ALHCALIOP

Corrected.

Lines 16-17, page 10: Quantitatively, 67

Lines 18-20, page 10: Considering all EPIC- CALIOP ALH pairs, 65

Corrected.

Line 21, page 10: observations of 675 nm AOD -> 675 nm AOD observations

Corrected.

Line 22, page 10: (Ichoku et al., 2002) -> Ichoku et al. (2002)

Corrected.

Lines 22-24, page 10: "but was modified to associate a subset of satellite retrievals within a 3 X 3 AOD subset centered at each site to a subset of 1-hour AERONET observations around EPIC scan time." It is not clear what the authors mean by 3 X 3 AOD subset. The sentence needs to be revised for clarity.

We revised the sentence into "but was modified to associate compare a subset of satellite retrievals within EPIC AOD retrievals over a 3×3 AOD subset pixels centered at each the AERONET sites to a subset of with 1-hour AERONET AOD observations around the EPIC scan time.".

Line 24, page 10: EPIC scan time -> the EPIC scan time

Corrected.

Lines 24-25, page 10: Comparison of EPIC AOD and AERONET are shown in Figure 10b. -> A comparison of EPIC and AERONET AODs is shown in Figure 10b.

Corrected.

Lines 25-26, page 10: The collocated AOD pairs, though with limited data samplings, have over 77

Corrected.

Line 26, page 10: EPIC AOD -> The EPIC AOD

Corrected.

Line 34, page 10: UV aerosol index -> the UV aerosol index

Corrected.

Line 1, page 11: because -> since

Corrected.

Line 9, page 11: both perform -> both of which obtain

Corrected.

Line 10, page 11: leave a space between the citations; Is it Omar et al or Torres et al?

Done. Double checked, and it is Omar et al.

Line 14, page 11: which are in contrast to clouds which-> which are in contrast to clouds that

Corrected.

Line 15, page 11: Because -> Since

Corrected.

Line 16, page 11: correlation -> the correlation

Corrected.

Lines 17-18, page 11: may results in a value UVAI from less than 1 to about 4 -> may result in UVAI values ranging from less than 1 to about 4

Corrected.

Line 19, page 11: EPIC's O2 bands -> the EPIC O2 bands

Corrected.

Lines 21-22, page 11: Based on our previous efforts in retrieving over-water dust ALH from the EPIC (Xu et al., 2017), we extend the retrieval algorithm to biomass burning

smoke aerosols over both the water and vegetated land surfaces. -> We extend our retrieval algorithm for retrieving over-water dust ALH from EPIC (Xu et al., 2017) to biomass burning smoke aerosols over both water and vegetated land surfaces.

Corrected.

Line 23, page 11: flexible spectral fittings that account for specifics of-> flexible spectral fitting that accounts for the specifics of

Corrected.

Line 25, page 11: then uses -> and then uses

Corrected.

Line 28, page 11: And, surface reflectance -> Surface reflectance

Corrected.

Lines 31-32, page 11: We found the algorithm captures AOD and ALH multiple times daily over both the water and vegetated land surfaces. -> The algorithm is able to retrieve AOD and ALH multiple times daily over both water and vegetated land surfaces.

Corrected.

Lines 33-34, page 11: , showing EPIC retrieved ALH has a rmse of 0.58 km and captures 52

Corrected.

Line 1, page 12: mrse -> rmse

Corrected.

Lines 1-2, page 12: and over 77

Corrected.

Line 2, page 12: What does an error envelope of +/- (0.05 + 10

We changed the "an uncertainty envelope of ± (0.05 + 0.1AOD)".

Line 4, page 12: the EPIC's UV bands -> the EPIC UV bands

Corrected.

Line 9, page 12: dust or smoke -> (dust or smoke)

Corrected.

Line 15, page 12: NASA the DSCOVR Earth Science Algorithms Program -> the NASA DSCOVR Earth Science Algorithms Program

Corrected.

Line 16, page 12: Office of Naval Research (ONR's) -> the Office of Naval Research (ONR)

Corrected.

Line 17, page 12: under the award -> under award

Corrected.

Line 20, page 12: NASA's -> the NASA; AERONET program -> the AERONET program

Corrected.

Line 21: the AOD data -> AOD data

Corrected.

Table 2 caption: in constructing the LUTs -> for constructing the LUTs

Corrected.

Figure 1 caption: Change to: EPIC instrument filter response function (blue) and atmospheric spectral transmission 5 (orange). Panel (a) includes all ten EPIC bands, whereas panels (b) and (c) show zoom-ins for the 688-nm channel in the O2 B-band and the 764-nm channel in the O2 A-band, respectively. Here, the atmospheric transmission is simulated by the UNL-VRTM model, with a spectral step size and a spectral full width at half maximum of 0.02 nm.

Done.

Line 5, page 21: physical principal for -> physical principle of

Corrected.

Line 6, page 21: scattering of aerosol -> scattering by aerosol

Corrected.

Line 7, page 21: path way -> pathlength

Corrected.

Line 8, page 21: than in the lower-altitude aerosol -> than those scattered by the lower-altitude aerosol; less chance -> lower chance

Corrected.

Line 5, page 22: at the geometry of -> for the geometry

Corrected.

Figure 4 legend: Change "Green vegetations" to "Green vegetation surfaces"

Corrected.

Figure 5: What does 3X3 aggregation mean? Do you aggregate 9 pixels at a time? Why? Some explanation is needed in the text and better wording in the Figure.

In the figure, we changed "3X3 aggregation" to "3x3 pixel aggregation", and changed "3x3 averaging" to "averaging over the 3x3 pixels"

Line 5, page 25: the statistics -> statistics

Corrected.

Line 6, page 25: red dot line -> red dotted lines; their respective -> the respective

Corrected.

Lines 7-8, page 25: reflectance at each EPIC band versus reflectance at corresponding MODIS bands -> reflectance in each EPIC band versus reflectance in the corresponding MODIS bands

Corrected.

Line 4, page 26: UTC time -> UTC times

Corrected.

Line 6, page 26: CALIOP sub-orbital track with an overpass time 19:05 UTC -> the CALIOP sub-orbital track with an overpass time of 19:05 UTC

Corrected.

Line 8, page 26: EPIC scan time -> the EPIC scan time

Corrected.

Line 5, page 27: UTC time -> UTC times

Corrected.

Line 6, page 27: CALIOP overpass time was at 18:15. -> The CALIOP overpass was at 18:15 UTC.

Corrected.

Line 4, page 28: Comparison of ALH retrieved from EPIC and the ALH derived from CALIOP level-2 aerosol extinction profile -> Comparison of ALH retrieved from EPIC and CALIOP level-2 aerosol extinction profile

Corrected.

Line 5-6, page 28: CALIOP orbital tracks are marked on EPIC RGB images in Figure 7–8. -> The CALIOP orbital tracks are marked on EPIC RGB images in Figures 7–8.

Corrected.

Line 7-8, page 28: Error bar of EPIC ALH represents standard deviation for an array of 3x3 24-km retrieval pixels, while the error bar of CALIOP ALH represents standard deviation of over 5 adjacent CALIOP 5-km footprints. -> The error bar for EPIC ALH represents the standard deviation for an array of 3x3 24-km retrieval pixels, while that for CALIOP ALH represents the standard deviation of over 5 adjacent CALIOP 5-km footprints.

Done.

Line 4, page 29: counterparts from -> corresponding Line 5, page 29: Color of -> The color of

Corrected.

Line 6, page 29: scatter -> scatter point; EPIC 680-nm AOD value of -> the EPIC 680-nm value for

Corrected.

Line 7, page 29: Dotted lines -> The dotted lines Line 8, page 29: one-by-one -> the one-to-one

Corrected.

Line 9, page 29: regression fitting -> regression fit; scatters -> scatter points; the linear -> linear

Corrected.

Line 10, page 29: scatters -> scatter points

Corrected.

Line 4, page 30: UVAI were -> UVAI was

Corrected.

Line 4, page 31: linear regression fitting -> the linear regression fit

Corrected.

Line 5, page 31: scatters -> scatter points

Corrected.

---

## Author Response (AR2)

**Reply to editor**

Dear Prof. Schmidt,

We thank you for your comments. We have addressed your comments in the manuscript, and we believe the manuscript is clearer in terms of these two questions. Below is a response to each of your comments. Your original comments are in black, and our replies are in blue.

1) The explanation regarding the insensitivity to BRDF (reviewer 3) is somewhat short. Can you provide a little more information and preliminary explanation? It is understood that this will be studied in more detail later, of course.

We added some more information about the EPIC observation geometry to explain the insensitivity to BRDF. Note, EPIC is parked at L-1 point and its viewing geometry is nearly constant (close to the 180 degree backscattering). Hence, once the surface reflectance database is well defined at the backscattering direction for the aerosol retrieval at one time, they are suitable to be used for other times. Now the text in section 3.3 becomes:

"It should be noted that the effect of non-Lambertian surface reflection may bias the ALH retrieval, because uncertainty in surface reflectance can substantially affect the ALH retrieval accuracy (see Appendix A). Nevertheless, this type of impact could be limited as EPIC's earth observations are confined within an almost constant viewing geometry with scattering angles between $165° – 178°$. Further studies are needed to examine the detailed impacts, which will be one of our future efforts."

2) It makes sense that the retrieval should be rather insensitive to SSA, but could you add a somewhat physical explanation, if possible?

We added more explanation for the sensitivity to SSA in the Appendix, which reads:

"DAOS ratios are sensitive to SSA to some degree, especially for large AOD values (Figure A1e). However, the sensitivity to SSA is much less overwhelmed than the sensitivity to AOD and surface reflectance because the reflectance at TOA depends more on surface reflectance and AOD (than SSA in relative sense). As a result, SSA only has marginal impact to the ALH retrieval error (green curves in Figure A2c-d), which is consistent with findings by Sanders et al. (2015)."